# Structural organization of p62 filaments and the cellular ultrastructure of calcium-rich p62-enwrapped lipid droplet cargo

Sabrina Berkamp [1,12] ✉, Lisa Jungbluth [1,12], Alexandros Katranidis [1,12], Siavash Mostafavi [1], Olivera Korculanin[1,2], Peng-Han Lu[3], Lotte Ickert[4], Maya M. Dierig [5], Lokesh Sharma [6], Lipi Thukral [6,7], Pitter F. Huesgen [5,8,9], Natalia L. Kononenko [4,10], Jörg Fitter[1,2], Rafal E. Dunin-Borkowski [3] & Carsten Sachse [1,11] ✉

The selective autophagy receptor p62/SQSTM1 is known to form higher-order filaments in vitro and to undergo liquid-liquid phase separation when mixed with poly-ubiquitin. Here, we determine the full-length cryo-EM structure of p62 and elucidate a structured double helical filament scaffold composed of the PB1-domain associated with the flexible C-terminal part and the solvent-accessible major groove. At different pH values and upon binding to soluble LC3, LC3-conjugated membranes and poly-ubiquitin, we observe p62 filament re-arrangements in the form of structural unwinding, disassembly, lateral association and bundling, respectively. In the cellular environment, under conditions of ATG5 knockdown leading to stalled autophagy, we imaged high-contrast layers consisting of p62 oligomers enwrapping lipid droplets by cryogenic electron tomography in situ, which we identified as calcium as well as phosphorus by compositional spectroscopy analysis. Together, we visualize the cellular ultrastructure of p62 oligomers with high calcium content as a potential early stage of autophagy.

Autophagy is a conserved cellular process that degrades and recycles cytoplasmic material such as aggregated proteins, damaged organelles and pathogens to maintain intracellular homeostasis[1,2]. While autophagy operates at a basal level for routine maintenance, it can also be activated by stress factors like nutrient deprivation, hypoxia and DNA damage[3]. There are three main autophagy pathways: macroautophagy, microautophagy and chaperone-mediated autophagy. In macro-autophagy (from here on simply referred to as autophagy), an autophagosome forms within the cytoplasm to enclose cellular material that is subsequently degraded and recycled after fusing with a lysosome. A series of defined autophagy core machinery complexes mediate the progression of cellular autophagy: Atg1/ULK1 kinase, PI-3 kinase, ATG9

[1]Ernst-Ruska Centre for Microscopy and Spectroscopy with Electrons, ER-C-3/Structural Biology, Forschungszentrum Jülich, Jülich, Germany. [2]RWTH Aachen, I.Physikalisches Institut (IA), Aachen, Germany. [3]Ernst-Ruska Centre for Microscopy and Spectroscopy with Electrons, ER-C-1/Physics of Nanoscale Systems, Forschungszentrum Jülich, Jülich, Germany. [4]CECAD Excellence Center & Center for Physiology and Pathophysiology, Faculty of Medicine and University Hospital Cologne, Cologne, Germany. [5]Institute of Biology II, University of Freiburg, Schänzlestraße 1, Freiburg, Germany. [6]Computational Structural Biology Lab, CSIR-Institute of Genomics and Integrative Biology, Mathura Road, New Delhi, India. [7]Academy of Scientific and Innovative Research (AcSIR), Ghaziabad, India. [8]CIBSS-Centre for Integrative Biological Signaling Studies, University of Freiburg, Schänzlestraße 1, Freiburg, Germany. [9]Central Institute for Engineering, Electronics and Analytics, ZEA-3, Forschungszentrum Jülich, Jülich, Germany. [10]Center for Molecular Medicine Cologne, Faculty of Medicine and University Hospital Cologne, University of Cologne, Cologne, Germany. [11]Department of Biology, Heinrich Heine University, Universitätsstr. 1, Düsseldorf, Germany. [12]These authors contributed equally: Sabrina Berkamp, Lisa Jungbluth, Alexandros Katranidis. ✉e-mail: s.berkamp@fz-juelich.de; c.sachse@fz-juelich.de

lipid scramblase, the ATG2-ATG18 lipid transfer complex and two ubiquitin-like conjugation systems[4]. The recognition of cargo can proceed in a non-selective or in a selective manner by either enclosing the bulk stochastically or through specific binding by selective autophagy receptors, respectively[5].

Selective autophagy requires cargo receptors that recognize poly-ubiquitinated cargo and interact with Atg8 family proteins[6,7]. The Atg8 family contains ubiquitin-like proteins: Atg8 in yeast and microtubule-associated protein 1 light chain 3 (LC3), as well as gamma-aminobutyric acid type A receptor-associated protein (GABARAP) in mammals[8]. The conjugation of Atg8 proteins to the lipid membrane is essential for phagophore membrane expansion[1] and is mediated by two autophagy-specific ubiquitin-like conjugation systems: ATG12-ATG5 and the LC3 conjugation system[9]. Selective autophagy receptors, like p62/SQSTM1 (from here on: p62)[10], contain an LC3-interacting region (LIR) motif that facilitates interaction with Atg8 proteins and a ubiquitin-binding domain[11,12]. At the early stages of autophagy, selective autophagy receptors like p62 thereby bridge ubiquitinated cargo via conjugated Atg8 to the forming autophagosome. Thus far, a series of additional autophagy receptors have been identified with LIR motifs as well as ubiquitin binding abilities in order to confer specificity and redundancy in the cargo recognition and removal[13]. For instance, for mitophagy, NBR1, optineurin, NDP52 and TAXBP1 have been characterized in addition to p62[14]. While p62 knock-outs are viable, increased p62 levels have been found to provide long-term benefits and promote proteostasis and longevity in response to stress in C.elegans[15].

p62 has a key role in selective autophagy by recognizing poly-ubiquitinated cargo and delivering it to the autophagosome[10]. Originally, p62 was described as an interaction partner for tyrosine-protein kinase Lck[16], while it was also shown to be involved in multiple pathways, including Wnt, Nrf2, mTOR and NF-κB signaling[17]. In the NF-κB pathway, p62 can interact with MEKK3, MEK5, aPKCs[18] and tumor necrosis factor receptor-associated factor 6 (TRAF6) as a mediator in tumorigenesis[19]. Moreover, p62 actively promotes the spatio-temporal organization of autophagy downstream effectors such as FIP200 at the cargo site[20]. p62 is a multi-domain protein with a Phox1 and Bem1p (PB1) domain, followed by a ZZ-domain and an intrinsically disordered region (IDR) up to the C-terminal ubiquitin-binding associated (UBA) domain (Fig. 1a). The PB1 domain (1–102) is responsible for p62 polymerization into long flexible filaments[21,22], and the short region 103–122 linking PB1 and ZZ domain has been shown to support polymerization[21] further supported by the formation of disulfide conjugates[23]. The PB1 domain was shown to be critical for its function in autophagy, as polymerization-deficient mutations excluded p62 from the autophagosome formation site[24]. The IDR contains several relevant interaction motifs, including the LIR and KEAP1-interacting region (KIR) motif[25]. The C-terminal UBA domain captures poly-ubiquitinated cargo[26]. The autophagy receptor NBR1 is structurally the most similar to p62 amongst the autophagy receptors, as it is closely evolutionarily related and directly interacts with p62 via its PB1 domain, albeit it is unable to form higher-order oligomers or filaments[22,27].

p62 is a well-studied cargo receptor that recognizes and removes various cellular cargoes: aggregated proteins in aggrephagy[10,28], damaged peroxisomes in pexophagy[29] and lipid droplets in lipophagy[30,31]. For lipophagy, the size of the removed lipid droplets is smaller than one μm, while larger lipid droplets were shown to be predominantly degraded by lipolysis[32]. At the same time, lipid droplets are a special cargo in the context of autophagy as they can also serve as a direct or indirect lipid source for the growing phagophore mediated by direct contact in addition to the ATG9A vesicles or ATG2 lipid bridges[33–36]. Lipid droplets were also found to serve as direct substrates for non-canonical autophagy during prolonged starvation after LC3-conjugation[37].

p62 has also been shown to undergo liquid-liquid phase separation[38] induced by poly-ubiquitin interaction with multiple polymerized p62 entities in vitro[39,40]. However, in vivo, p62 phase-separated condensates recruit many autophagy regulators that can affect the mobility of the separated phase and, as a result, autophagy efficiency. For instance, Keap1 over-expression was observed to increase the rigidity of p62 condensates, thus promoting autophagy[41]. NBR1 enhances p62 condensate formation and poly-ubiquitin affinity[42]. In addition to binding partners, the oligomeric state of p62 and post-translational modifications can affect the phase separation properties, e.g., as the IDR of p62 harbors the KIR motif that, when post-translationally modified, has been shown to enhance the interaction with Keap1[43]. The quaternary structural organization of p62 presents another level of regulation for cellular phase separation, as phosphorylation of the PB1 domain by PKA was identified and shown to disrupt PB1 domain interactions[44].

In order to improve our detailed structural understanding of purified p62, we determined the cryo-electron microscopy (cryo-EM) structure of full-length p62. The structure revealed the double helical organization of an N-terminal PB1-domain scaffold, while the more poorly resolved partially disordered C-terminal part resided in the lumen and the major groove of the assembly. Binding studies of LC3b and poly-ubiquitin showed that p62 filaments can undergo structural re-arrangements of disassembly or filament bundling, respectively. When p62 filaments were added to LC3b-conjugated membranes, they associated laterally and also unwound their regular helical structures into bubbles at physiological pH. Native ultrastructural imaging of p62-positive structures using cryo-electron tomography (cryo-ET) in ATG5-depleted RPE1 cells revealed stalled intermediate autophagy stages of p62-enwrapped lipid droplets. Interestingly, the surrounding p62 layer had a dark contrast and contained high levels of calcium and phosphorus based on energy-dispersive X-ray spectroscopy analysis (EDX). Proximity proteomics experiments showed nearby ER-resident calcium exporters. The high-resolution visualization of p62-enwrapped cargo together with calcium may present an important intermediate state of autophagy initiation.

## Results

### p62 forms a double-helical filament with a PB1 scaffold and a flexible C-terminus in the major groove

Previously, we determined the three-dimensional (3D) cryo-EM structure of the p62-PB1 domain arranged in a series of different filamentous assemblies[22]. In order to advance our structural understanding of full-length human p62, we optimized the previous preparations of full-length p62 for improved homogeneity. As previous studies suggested an effect of pH on the assembly conditions[45], we compared the preparations at different pHs ranging from 6 to 8 (Supplementary Fig. 1a). Classifications of small cryo-EM data sets revealed that at pH 6.0 the majority of the filament classes were helically regular while at pH 8.0 a large fraction of them were found unwound resembling an open bubble (Supplementary Fig. 1b). Therefore, we recorded a full cryo-EM data set at pH 6.0 revealing long and flexible filaments with a width of approximately 15 nm (Fig. 1b). Using segmented helical reconstruction[46–48], we set out to determine the structure of p62 filaments. Due to pitch heterogeneity between 130 and 160 Å (Fig. 1c, Supplementary Fig. 1c), we refined a more homogeneous subset with a 150 Å pitch and 13.6 units per turn (corresponding to 11.0 Å helical rise and a −26.1° helical rotation), including dihedral symmetry and obtained a 3D reconstruction at 4.5 Å global resolution (Supplementary Fig. 1d, e, Table 1). Consistent with the class averages, the local resolution varied significantly from a well-defined PB1 domain scaffold at 4.2 Å resolution up to over 9 Å for the more flexible regions (Fig. 1d). The cryo-EM structure revealed a left-handed double-helical filament with two antiparallel strands of PB1

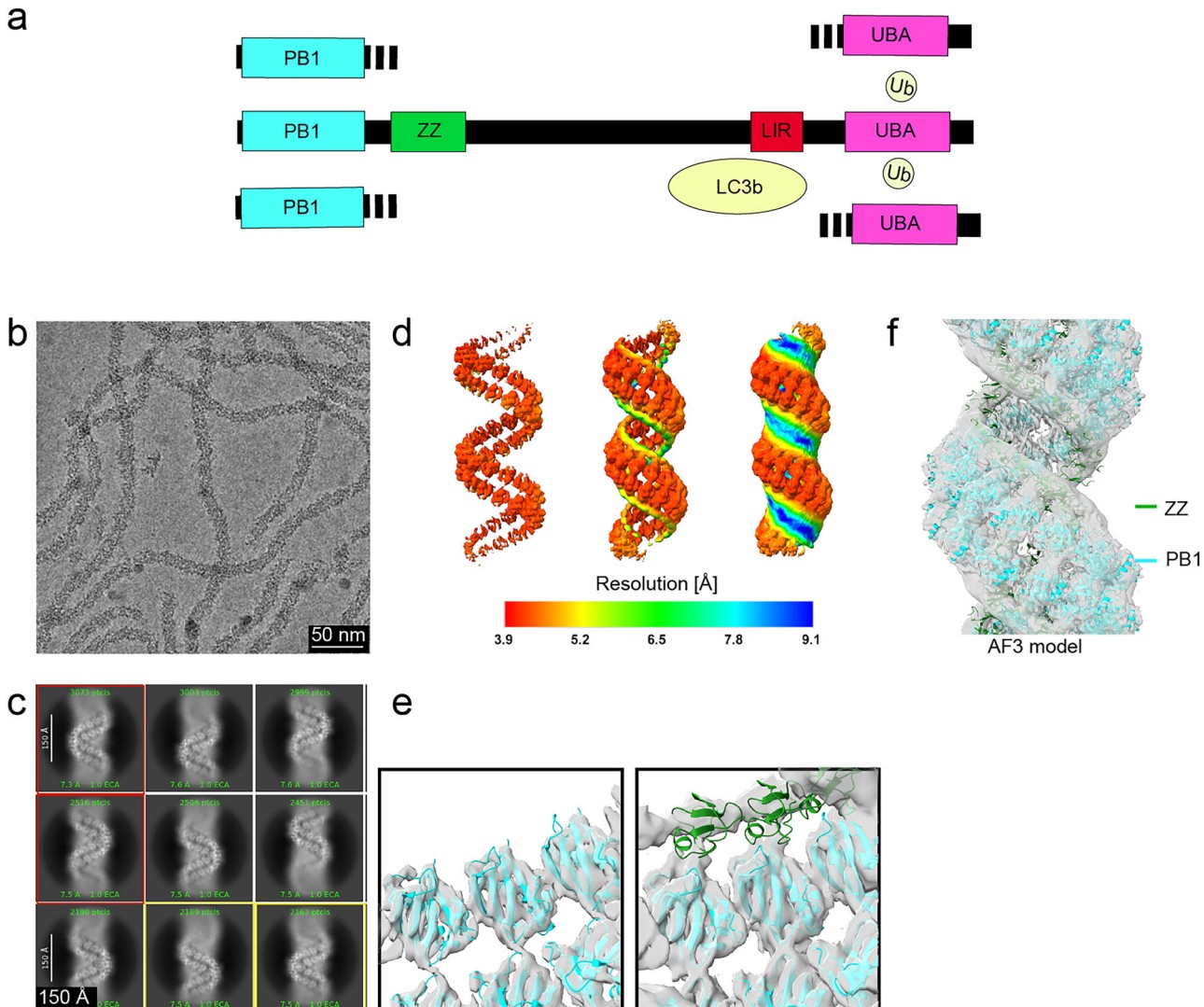

**Fig. 1 | Cryo-EM structure and atomic model of full-length p62 filaments. a** p62 domain plot including main interaction sites: the N-terminal PB1 domain responsible for self-polymerization, followed by the zinc finger ZZ domain, a long region of predicted disorder including the LC3-interacting region (LIR) motif and the C-terminal UBA domain recognizing ubiquitinylated cargo. UBA domains of neighboring p62 proteins can interact with each other through the bound polyubiquitin. **b** Representative cryo-micrograph (out of 4038) with long and flexible p62 filaments. **c** 2D classes showing well-defined densities of the filamentous scaffold in addition to blur, including apparent variation in pitch, with higher helical pitch classes (red frame) and lower helical pitch classes (yellow frame). **d** Three-dimensional density of p62 colored according to local resolution values showing PB1 domains arranged in a left-handed double helical architecture. The map rendered at three different thresholds (left, center, right) shows that the PB1 domain is well-resolved (~4.5 Å resolution), the ZZ domain appears as a featureless blob at ~6.5 Å resolution, and the remaining C-terminal domains are not resolved. **e** Left: The atomic model of the PB1 domain fits well into the well-defined PB1 density at a higher threshold. Right: The atomic model of the ZZ domain of p62 fits in the featureless blobs seen at a lower threshold. **f** The atomic model of the helical p62 filament with highlighted PB1 (cyan) and ZZ (green) domains fitted into the filament density displayed at a low threshold. The experiment on the cryo-EM for structure determination was conducted once ($N = 1$), and 4038 micrographs were collected.

domains in the full-length p62 assembly. The atomic model of the PB1 domain of p62 (PDB: 6TGY) fitted well in the best-defined scaffold density supporting β-strand separation (Fig. 1e left). At lower density thresholds, the ZZ domain envelope emerged connected to the PB1 domain, albeit at lower resolution (Fig. 1e right). The densities of the associated C-terminal domains were weak and poorly resolved as they did not follow helical symmetry, presumably due to molecular flexibility with respect to the PB1 strands, which is in agreement with the assigned structurally disordered region in residues 169–338[21]. Based on our p62 filament structure, we expanded the domains to match the PB1 density to the experimentally observed helical symmetry assembly (Fig. 1f). The resulting p62 double helix architecture is built on the scaffold of the PB1 domain, generating a major groove that can accommodate the flexible C-terminal part.

## p62 filament interactions with LC3b and ubiquitin

To experimentally study the interactions of p62 filaments with LC3b and ubiquitin, we employed single-particle fluorescence microscopy in total internal reflection fluorescence (TIRF) mode and negative staining EM. We used the above-characterized preparations of p62 filaments at pH 8 as they were closer to physiological pH than the stable pH 6 filaments formed for structural analysis. Subsequently, we labeled them at lysine residues with multiple Cy5 dyes (p62 filaments-Cy5) while cyan fluorescent protein (CFP) was fused to LC3b (CFP-LC3b). Consistent with the observed p62 filaments, elongated p62 structures were visible in the Cy5 channel, rather mobile and diffusing above the surface in dilute nM concentrations on the cover slip (Fig. 2a, Supplementary Movie 1). Upon addition of equimolar amounts of CFP-LC3b, the same filament became visible in the CFP channel as well

**Table 1 | Cryo-EM structure determination of full-length p62**

| Processing type | Helical segments |
|---|---|
| Microscope | Talos Arctica 200 keV |
| Detector | Gatan K3 |
| Magnification | 100,000x |
| Defocus range [μm] | −0.5 to −3.0 |
| Physical pixel size [Å] | 1.6778 |
| Total dose [e⁻/Å²] | 50 |
| Micrographs | 4038 |
| Extraction box size [Å] | 503 |
| Overlap [%] | 90 |
| Initial no. of segments | 1,000,000 |
| Final no. of segments | 600,000 |
| Helical symmetry | |
| Pitch [Å] | 150 |
| Rise [Å] | 10.985 |
| Rotation [°] | −26.141 |
| Global map resolution (FSC = 0.143)/FDR-FSC [Å] | 4.5/4.5 |

(Fig. 2b, Supplementary Movie 2), supporting that LC3b was bound to the p62 filaments. Upon addition of GST-4xUbiquitin (GST-4xUb) to the labeled filaments, we observed a rapid and spontaneous formation of μm-sized condensates that displayed minimal mobility (Fig. 2c, Supplementary Movie 3). Interestingly, the subsequent addition of CFP-LC3b to the formed condensates resulted in the appearance of smaller mobile diffusing fragments (Fig. 2d, Supplementary Movie 4).

The corresponding p62 filament samples in nM concentration were also imaged by negative staining EM. In the presence of CFP-LC3b, p62 filaments exhibited a distinctive decoration in comparison with the filaments alone (Fig. 2e, f, white arrows in insets). A quantitative analysis of the filament length distribution revealed a shift towards shorter filaments for CFP-LC3b (Fig. 2g), suggesting that interaction with LC3b induced depolymerization of p62 filaments. Similarly, in the presence of CFP-LC3b, GST-4xUb-induced condensates were disintegrated into smaller substructures (Fig. 2h, i). In support of previous studies[49,50], our single-particle studies in fluorescence and electron microscopy performed in diluted nM concentration showed that poly-ubiquitinated cargoes induce condensate formation, while LC3b can promote disassembly of preformed p62 filaments as well as preformed condensates.

In order to molecularly assess the cryo-EM structure and the presented binding experiments, we generated a multimer AI-prediction of full-length p62 using AlphaFold3 (AF3)[51], consisting of ten subunits that formed closed rings mediated by the well-characterized PB1 domain interactions (Supplementary Fig. 2a, Materials and Methods). In an expanded model matching the helical symmetry, the C-terminal UBA domain fits tightly to the major groove next to the PB1 domain that is consistent with the cryo-EM density (Supplementary Fig. 2b). When the model is seen in top view, the PB1 domain scaffold and UBA domain are located at the outer rim of the filament while the ZZ domain is located on the inside. Notably, parts of the IDR, including the LIR and KIR binding motifs, are positioned towards the filament's outside, supporting the observed molecular access of p62 interacting partners through the major groove of the filament. To assess the interaction accessibility within the p62 filament in more detail, we generated a second p62 multimer model in the above-described way, this time with LC3b known to bind to the LIR motif of p62[28]. In the resulting model, LC3b is tightly fit in the filament's major groove surface between the PB1 domain and covering the UBA domain (Supplementary Fig. 2c). The LIR of p62 is positioned outward and bound in the hydrophobic pocket 1 (HP1) and 2 (HP2) of LC3 in

agreement with previously determined X-ray structures[52]. A third p62 AF3-model with bound ubiquitin revealed that ubiquitin also resided in the major groove, suggesting a competing mode between LC3b and ubiquitin binding (Fig. 2j). Together, the AF3-based integrative molecular model of the p62 filament supports the observed accessibility to binding partners of ubiquitin and LC3b, while it also provides a molecular framework for the observed opposing higher-order structures of filament bundling and disassembly.

## p62 filament membrane interactions

In order to more closely mimic the cellular conditions, we performed the interaction studies at higher μM concentrations of p62 and investigated the binding of GST-4xUb and LC3b maleimide-conjugated to small unilamellar vesicles (SUVs). Subsequently, we visualized the natively preserved ultrastructures using cryo-ET. To better characterize the fine ultrastructure of p62 filaments, we recorded tomograms using the Volta phase plate for detailed high-contrast visualization. The resulting 3D tomograms showed 15 nm-wide flexible p62 filaments consistent with repeating pitches of 14 nm along the filament exhibiting regular major groove indentations matching the pH 6 conditions described above (Fig. 3a). Closer inspection revealed overlapping filaments crossing underneath each other, branching filaments where individual PB1 scaffold strands are adopting a close to 90° bent as well as a cross-over reminiscent of Holliday junctions in double-stranded DNA. Due to the underfocus-induced white halo of the phase plate surrounding the filament density, we turned back to conventional cryo-imaging at underfocus and closer to physiological pH 7.4 for the tomograms matching the binding experiments above. Upon addition of GST-4xUb, p62 filaments were found to concentrate locally by forming bundles of parallel filaments supported by filament image segmentations (Fig. 3b). Bundle formation was accompanied by the formation of μm-sized 3D agglomerates that were often too thick to be imaged by cryo-ET while visible by negative staining EM (Supplementary Fig. 3) being consistent with the previous in vitro observations of phase separated p62 bodies[49,50]. When we incubated p62 filaments with LC3b-conjugated SUVs, we observed unbound filaments as in the p62-only control in proximity to SUVs as well as SUV bound filaments (Fig. 3c). When not bound, the p62 exhibited the typical filament appearance with a regular 15 nm pitch and major groove indentations as well as unwound stretches of the high pH 8 p62 sample alone (Fig. 3d). When p62 filaments were found connected to the SUV density, they were either laterally associated along the membrane of the SUVs or bound through their ends. As in solution, some laterally associated filaments were characterized by densities giving up the regular helical pitch 15 nm repeat, giving rise to more open and unwound stretches. The co-incubation of GST-4xUb and LC3b-conjugated SUVs resulted in less dense network of filament bundles in comparison with GST-4xUb alone as well as SUV bound filaments (Fig. 3e). Together, the cryo-ET visualization of purified p62 revealed the flexible properties of p62 filaments including cross-overs and branching, bundling of parallel p62 filaments in GST-4xUb bound condensates as well as the lateral association of the helical assembly on the surface of LC3b-conjugated membranes.

## ATG5 depletion induces p62-positive lipid droplet accumulation in RPE1 cells, primary neurons and brain cortical tissues

In order to further our understanding of the structural organization of the autophagy receptor p62, we decided to visualize the cargo receptor in cells. Capturing rare and transient p62 punctae is challenging; therefore, we turned to a model system of ATG5 depletion that was shown to increase the size and number of p62 punctae[28,53] along with increased numbers of lipid droplets[54]. We stalled autophagy by treating human RPE1 cells stably expressing mCherry-p62 with siRNA against ATG5 for 72 h. To confirm successful ATG5 siRNA knockdown, we performed a Western blot of the RPE1 cells showing a

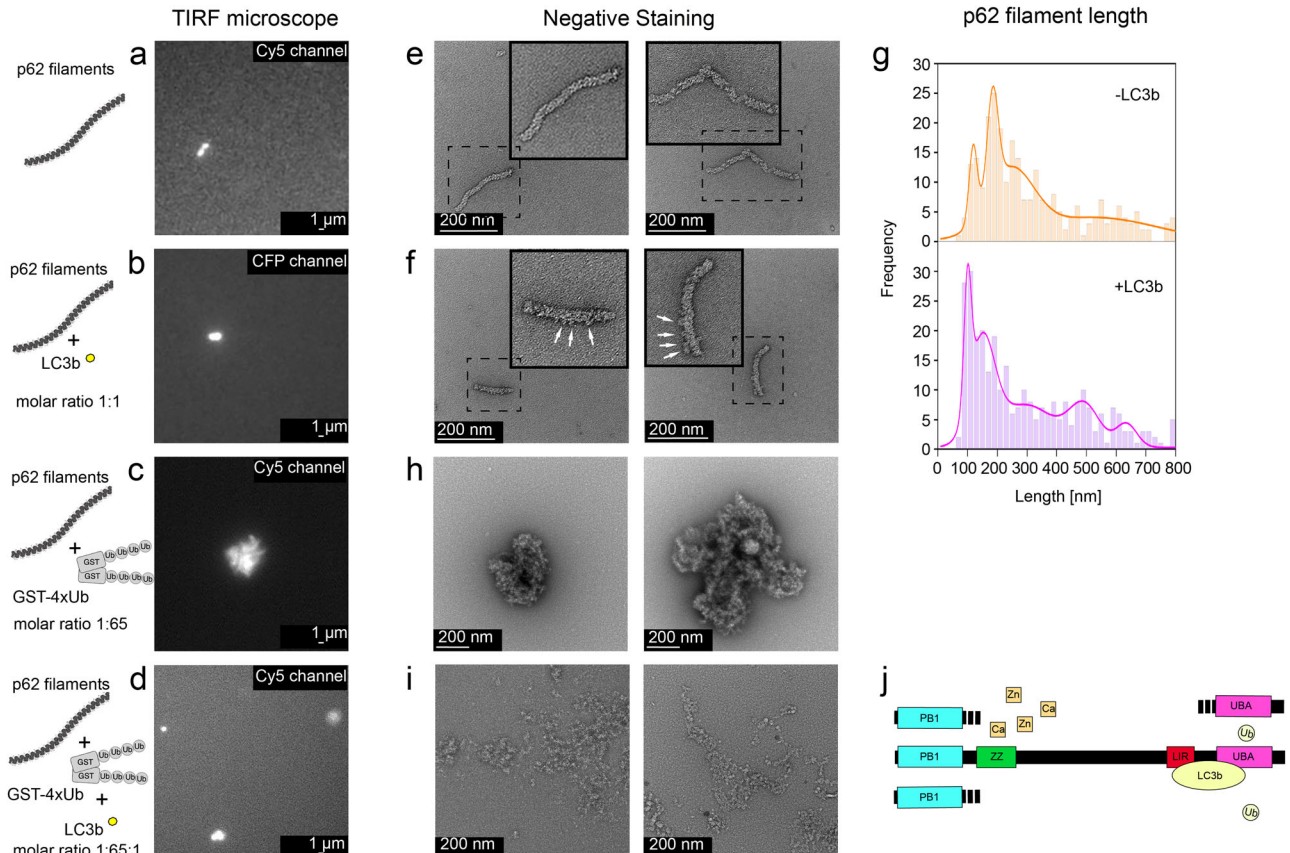

**Fig. 2 | LC3b and poly-ubiquitin interactions with p62 filaments visualized by single-particle fluorescence microscopy and negative staining EM.** p62 filament (**a**) labeled with Cy5 on a TIRF microscope and (**b**) in the presence of CFP-LC3b. **c** p62 condensates forming in the presence of GST-4xUbiquitin (GST-4xUb). **d** p62 condensates formed by GST-4xUb disintegrate after addition of LC3b. **e** Negative staining of p62 filaments. Dashed boxes correspond to an inset at higher magnification. **f** Negative staining of p62 filaments decorated with LC3b (sample as in B) (white arrows in inset). **g** Histograms of p62 filament length before and after addition of LC3b. **h** Negative staining of p62 condensates (sample as in D with GST-4xUb). **i** Negative staining of p62 condensates after addition of LC3b (sample as in D with GST-4xUb). **j** Illustration of revealed p62 domain interactions: The ZZ domain may potentially bind divalent cations $Zn^{2+}$ and $Ca^{2+}$, while interaction of LC3b via the LIR motif competes with the binding of ubiquitin to the UBA domain. All experiments were conducted three times ($N = 3$). Source data are provided as a Source Data file.

reduction in ATG5 expression levels by 90% in comparison with the control samples (Fig. 4a). To validate the phenotype upon ATG5 knockdown, we used confocal fluorescence microscopy to detect p62 punctae via mCherry-p62, as well as lipid droplets by using the lipid-droplet specific Lipi-Blue dye in conditions of no transfection, and transfection with ATG5 siRNA, respectively (Fig. 4b, c). As a control for our mCherry-p62 overexpression, we imaged lipid droplets in wildtype (WT) RPE1 reference cells with Lipi-Blue after no transfection, mock transfection and transfection with ATG5 siRNA using confocal microscopy (Supplementary Fig. 4a–c). These experiments confirmed that ATG5 knockdown in human RPE1 cells resulted in the accumulation of p62-positive punctae and an increase in lipid droplet content, establishing the conditions of a cellular model to study the spatial association between p62 and lipid cargo.

In order to verify that our observations were not limited to adherent RPE1 cells, we extended our experiments to mouse primary neurons and brain cortical tissue. First, we isolated primary neuronal cells from conditional *Atg5*flox:CAG-Cre[Tmx] newborn mice, a model previously shown to exhibit robust and complete knockout (KO) of ATG5[55]. After fixation, we labeled WT and KO astrocytes and neurons with an antibody against p62, and stained lipid droplets with Lipidspot488 (Fig. 4d–g). Second, we analyzed cortical brain sections from three-month-old WT and conditional ATG5 (cKO) mice, in which ATG5 deletion is driven by the neuronal-specific *Scl32a1* promoter[56]. These sections were stained with

antibodies against p62 and perilipin1 (PLIN1), a protein associated with the surface of lipid droplets (Fig. 4h, i). We used confocal microscopy to image these samples and quantified the amount of lipid droplets per cell, the amount of p62 punctae per cell, the average size of the lipid droplets and the percentage of lipid droplets that co-localized with p62 signal in each cell. Both in human and mouse cells, we observed an increase in the amount of lipid droplets, while this effect was most pronounced in mouse primary KO astrocytes and cortical 3-month-old KO neurons (Fig. 4j). Additionally, as expected, we found a significant increase in the number of p62 punctae upon ATG5 KD or KO in the analyzed cells (Fig. 4k). In the absence of ATG5, the average size of the lipid droplets increased from 0.3 to 0.4 µm², from 0.3 to 0.6 µm², from 0.7 to 0.8 µm², for mCherry-p62 RPE1, mouse astrocytes and neurons, respectively (Fig. 4l). The mouse cortical neurons were harder to quantify as they, on average, contained few, very large lipid droplets. Finally, we also detected a significant increase in the number of lipid droplets that co-localized with p62 punctae in those samples lacking ATG5 (Fig. 4m). Interestingly, in our mock-transfected RPE1 cells, we also observed a moderate increase in lipid droplet sizes (Supplementary Fig. 4d), possibly due to the fact that the cationic lipid mixtures used in transfection mixtures have been observed to induce autophagy, promoting LC3 conversion and p62 degradation[57,58]. Together with the autophagy-stalling effect of ATG5 siRNA, the lipids likely contribute to further enrichment of p62-positive structures. In conclusion, when transfecting RPE1 cells with

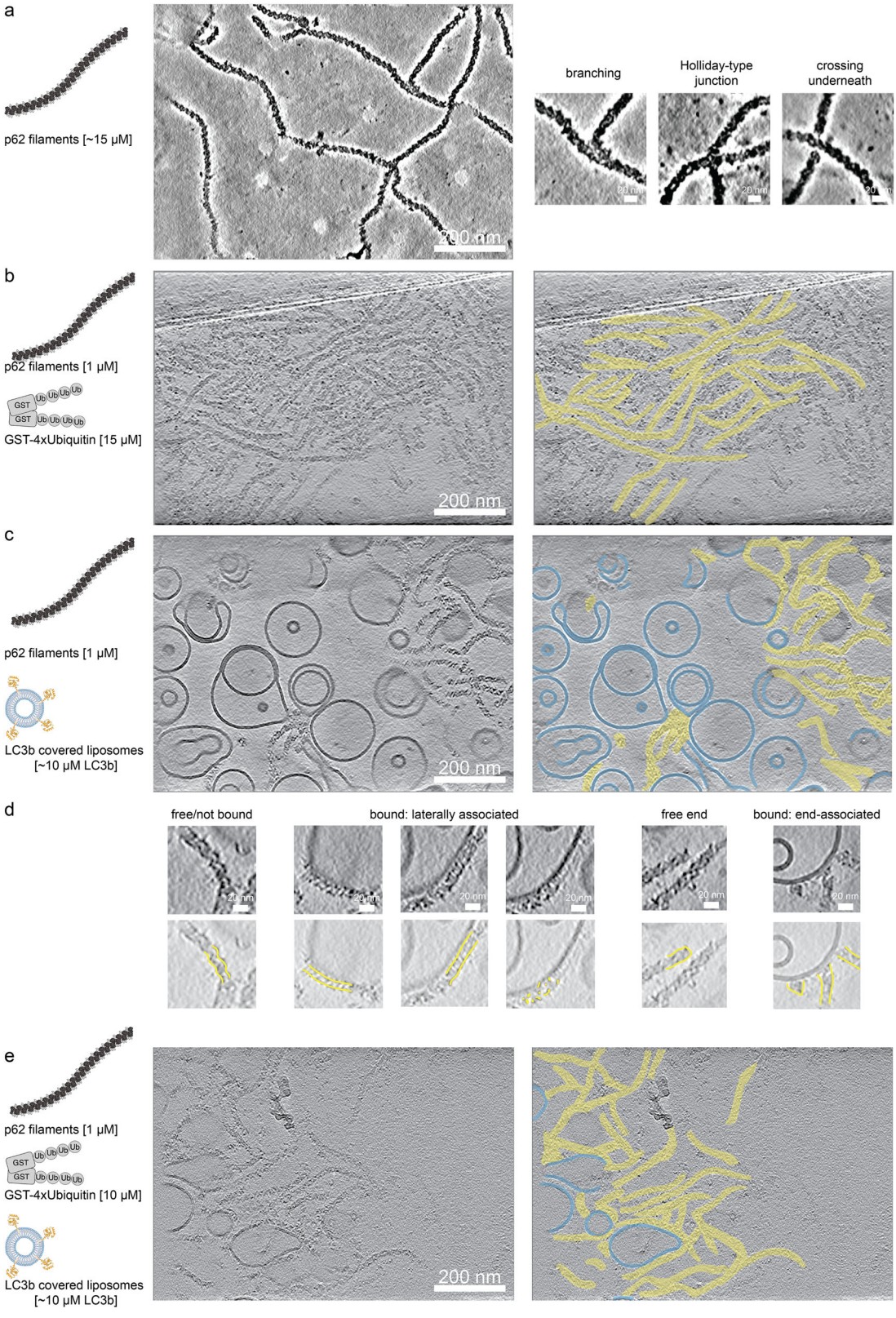

ATG5 siRNA, and upon ATG5 KO in mouse primary astrocytes, mouse primary neurons and/or 3-month-old mouse cortex, we observed an accumulation of lipid droplets co-localizing with p62 punctae, as well as a size increase for lipid droplets. The characterized condition of ATG5 knockdown in human RPE1 cells may be well suited for the in situ visualization of p62 punctae adjacent to lipid droplet cargo, as autophagosome formation and trafficking to the lysosome is impaired.

## p62 encapsulates lipid droplets by a discontinuous coat of variable thickness

Based on the established condition of accumulated co-localized p62 punctae and lipid droplets, we set out to perform detailed in situ structural analyses to study p62 during lipid droplet cargo recognition. RPE1 cells were grown under the same conditions described above and placed on micropatterned electron microscopy grids, vitrified,

**Fig. 3 | p62 filament interactions with LC3b-conjugated liposomes and poly-ubiquitin visualized by cryo-electron tomography. a** Left. Representative micrograph of p62 filaments recorded with Volta phase plate. Gold fiducials were computationally removed in the locations of the smooth circles ($N = 54$ tomograms). Right. Close-up insets showing the regular 15 nm pitch features along the filament. p62 filaments were found branching by 90° (left), cross-overs as in the Holliday junctions of DNA (center) and crossing each other underneath (leftright). **b** Left. Upon addition of GST-4xUbiquitin, the formation of parallel p62 filament bundles was observed ($N = 20$ tomograms). Right. Segmentations of closely packed p62 filaments aligned in parallel. **c** The addition of pre-formed p62 filaments to LC3b-conjugated small unilamellar vesicles (SUVs) showed bound and unbound p62 filaments ($N = 80$ tomograms). **d** Top. p62 filaments were found not-bound (left), bound laterally associated with the membrane (left center), free ends (right center) or through their ends (right). Bottom. Outline of p62 densities is highlighted in yellow. For laterally associated p62, regular helical and unwound assemblies were observed. **e** Co-incubation of GST-4xUbiquitin and LC3b-conjugated SUVs led to a dense network of p62 bundled filaments partially bound to SUVs ($N = 42$ tomograms).

subsequently thinned to ~150 nm using a focused ion beam (FIB) and visualized using a scanning electron microscope (SEM) (Fig. 5a). To verify that the lamella contained mCherry-p62, we imaged the sample using an integrated cryo-fluorescence microscope and correlated it with the FIB-SEM image[59] (Fig. 5b). When we visualized these p62 structures using super-resolution cryo-confocal microscopy, they showed circularly resolved mCherry-p62 signal on their rim suggesting the cargo encapsulation by the autophagy receptor p62 (Fig. 5c). At these sites, a total of 25 tomograms were recorded while those with green autofluorescence from lysosomes were not further analyzed. In the reconstructed tomograms, we identified characteristic structures made of a homogeneous texture of spherical shape suggestive of a lipid droplet[60] (Fig. 5d) in agreement with the frequent co-localization of p62 and lipid droplet punctae demonstrated above. A dark and highly dense layer was found surrounding the droplet in addition to a single phospholipid bilayer. The homogeneously textured spheres were similar in size measuring between 510–530 nm (Fig. 5e). The surrounding layer, or coat, varied in thickness within a single structure as well as between structures ranging between 4 and 27 nm, with a mean of 12.6 nm (Fig. 5f and Supplementary Fig. 5). The distance between the surface of the sphere and the surrounding single lipid membrane was relatively uniform at 13.8 nm with some weaker protein densities filling this space while other densities connecting these two structures (Fig. 5g, h). The segmented tomograms revealed that when the layer was thin, patches or islands of dark contrast were found distributed around the surface of the lipid droplet (Supplementary Fig. 6). However, when the layer was thick, it was completely continuous. Next to the lipid droplet, several small vesicles between 35 and 52 nm in diameter were observed, consistent with previously described ATG9A vesicles[35]. In summary, ATG5 knockdown in human RPE1 cells yielded p62-positive spherical structures of 500 nm diameter composed of an inner lipid droplet and a p62-containing, dense layer of variable thicknesses covering the surface, surrounded by a single bilayer membrane.

**Elemental analysis reveals calcium and phosphorus in the p62 positive layer**
Given the strong observed intensity of the surrounding p62-positive layer with respect to the gray levels of the lipid droplet and cytosol, we wondered whether non-organic elements contributed to the high contrast. To address this question, we employed scanning transmission electron microscopy (STEM) coupled with energy-dispersive X-ray (EDX) spectroscopy to record element-specific X-rays from the mCherry-p62 RPE1 lamella treated with ATG5 siRNA investigated above (Fig. 6a–d). For this sample, we acquired STEM images on a high-angle annular dark-field (HAADF) detector and simultaneously recorded the back-scattered EDX signal (Fig. 6e). The EDX signal was averaged over the excised regions of the cytosol, lipid droplet core and the p62 coat (Supplementary Fig. 7). For the cytosol, we detected mostly carbon (C), nitrogen (N) with low and oxygen (O) with high intensity at approx. 300, 400 and 500 eV, respectively. For the lipid droplet, we observed enriched C in comparison with the cytosol, reduced N and O, consistent with the presence of triacyl glycerides and sterols within the sphere of lipid droplets. In both regions, no inorganic elements were

observed at higher energies. For the p62 coat, we found the C, N, O signal comparable to the cytosol with clear additional inorganic signals of magnesium (Mg), calcium (Ca) and phosphorus (P) at 1250, 2000 and 3800 eV, respectively, supporting the observation of the relatively strong contrast in the raw tomograms. The detected high levels of Ca and P suggest the presence of calcium phosphate. As a control, we focused on lipid droplets showing no p62 co-localization and obtained a very similar EDX signal of the core lipid droplet region while lacking the high levels of Ca and P in their surrounding (Supplementary Fig. 8). Other p62-positive single-membrane vesicular structures resembling lysosomes showed no differences in EDX profiles between the vesicular area and the cytosol (Supplementary Fig. 9).

Given the unexpected finding of inorganic elements located in the p62 coat, we tested whether the addition of $Ca^{2+}$ ions had an effect on purified p62 filaments. When increasing the $CaCl_2$ concentrations from 100 nM to 100 mM, we observed a bundling of p62 filaments at 100 mM $CaCl_2$ in negative staining EM (Fig. 6f–h). Moreover, when testing other divalent cations such as $Zn^{2+}$ and $Mg^{2+}$, we found that the addition of $ZnCl_2$ had a stronger clustering effect already visible at 100 µM, while the addition of $MgCl_2$ did not show a significant bundling (Supplementary Fig. 10). Together, these experiments show that the observed p62-positive structures correspond to lipid droplets surrounded by a coat made of p62, including significant amounts of calcium (Fig. 6i) that also promotes the formation of p62 oligomer bundles in vitro and may explain the characteristic dense layer observed in the tomograms.

**Proximity proteomics identifies calcium transporters in proximity to the p62 encapsulated lipid droplets**
In order to provide further molecular insights into the observed p62-encapsulated lipid droplets, we turned to a proximity proteomics experiment[61]. In short, the miniTurbo enzyme was fused to mCherry-p62 and a stable RPE1 cell line was generated. Subsequently, proteins in the immediate proximity of p62 were biotinylated, pulled down and identified using mass spectrometry (Fig. 7a, b). The abundance of biotinylated proteins was estimated for the 72-h ATG5 knockdown (including mCherry-p62 and miniTurbo), i.e., T-mCh-p62 ATG5 KD, and compared with a control of endogenously expressed ATG5 (including mCherry-p62 and miniTurbo), i.e., T-mCh-p62, as well as a second control devoid of the miniTurbo enzyme (including mCherry-p62), i.e., mCh-p62. Western blot confirmed the absence of ATG5 in the T-mCh-p62 ATG5 KD sample, while ATG5 was detected in the mCh-p62 and T-mCh-p62 control samples (Fig. 7c). Consistently, we found neither ATG5 nor ATG16 in the proteomic data of the T-mCh-p62 ATG5 KD sample (ProteomeXchange-ID PXD066163). We detected several known interactors of p62 and other early autophagy proteins in both the T-mCh-p62 and T-mCh-p62 ATG5 KD samples: NBR1, ATG13, ZFYVE1/DFCP1 and TRAF6, while they were not present in the mCh-p62 control. Statistical analysis of relative peptide abundance between the T-mCh-p62 ATG5 KD and the T-mCh-p62 endogeneous ATG5 sample identified several other proteins related to autophagy that were only present in the absence of ATG5 (Fig. 7d). Most interestingly is the presence of SIRT1 and alpha-synuclein as these proteins have been described to be present on lipid droplets involved in lipophagy[62,63].

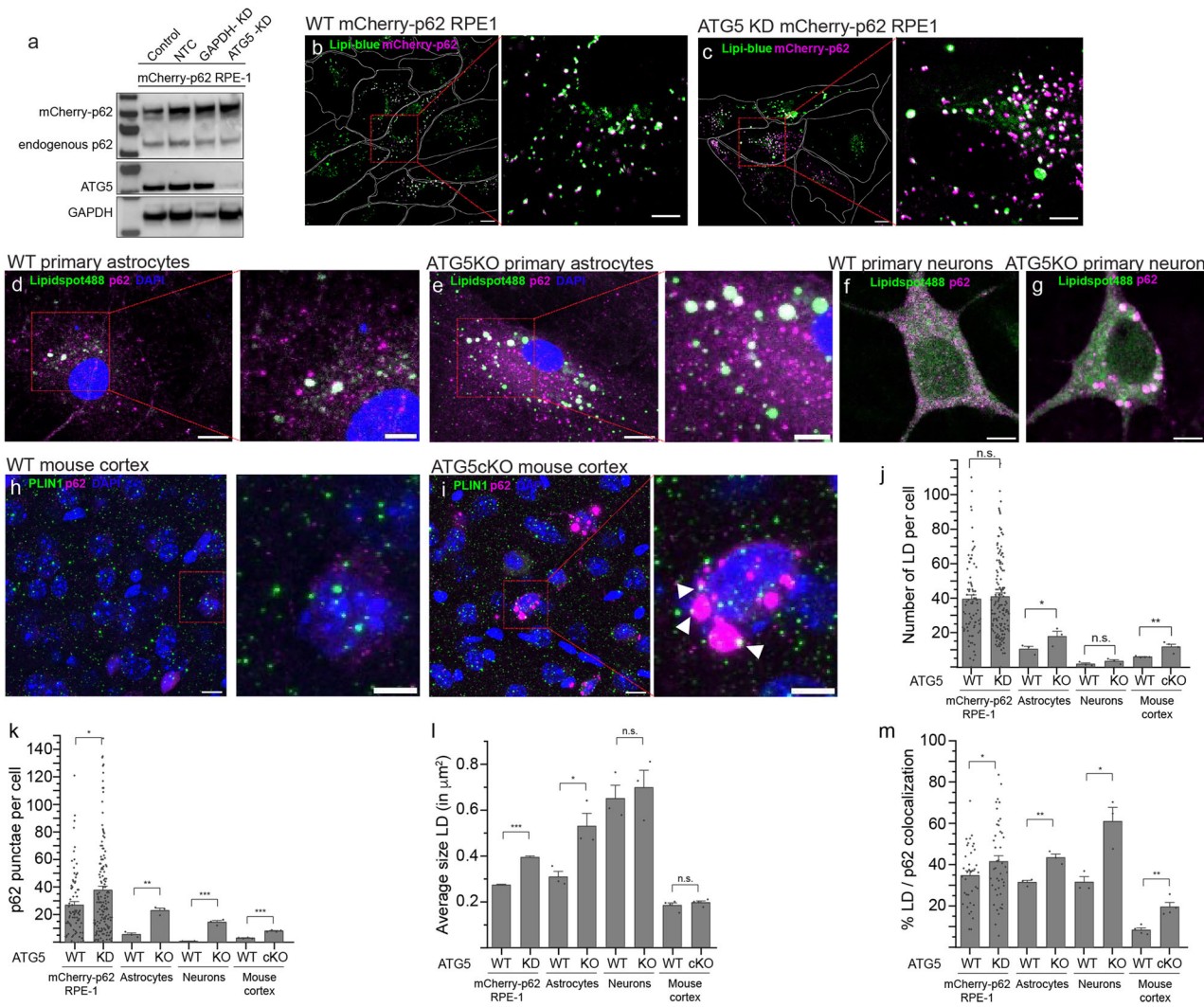

**Fig. 4 | Confocal fluorescence microscopy analysis of p62 punctae and lipid droplets upon ATG5 knockdown (KD) or knockout (KO). a** Western blot confirming the knockdown of ATG5 expression, including a control knockdown of GAPDH in mCherry-p62 RPE1 cells. Human mCherry-p62 RPE1 cells stained with Lipi-Blue for 24 h at conditions of (**b**) no transfection, and (**c**) 72 h after a transfection with ATG5 siRNA. Mouse Atg5flox:CAG-Cre^Tmx astrocytes stained with Lipidspot488 and anti-p62 antibodies 14 days after (**d**) control (i.e., EtOH) treatment and (**e**) tamoxifen treatment to induce ATG5 KO. Mouse Atg5flox:CAG-Cre^Tmx neurons stained with Lipidspot488 and anti-p62 antibodies 14 days after (f) (i.e., EtOH) treatment, (**g**) tamoxifen treatment to induce ATG5 KO. Sections of 3-month-old mouse cortex, stained with antibodies against p62 and perilipin1, (**h**) WT mice and (**i**) ATG5 cKO littermates. Quantifications for the displayed samples: (**j**) number of lipid droplets per cell. Statistical tests between the RPE1 samples showed $p = 0.687$, for the astrocyte samples it showed $p = 0.098$, for the neurons it showed $p = 0.225$ and for the mouse cortex it showed $p = 0.00685$. **k** Number of p62 punctae per cell. Statistical tests between the RPE1 samples showed $p = 0.0109$, for the astrocytes it showed $p = 9.87 \times 10^{-4}$, for the neurons it showed $p = 3.00 \times 10^{-4}$ and for the mouse cortex it showed $p = 3.59 \times 10^{-6}$. **l** average size of lipid droplets. Statistical tests between the RPE1 samples showed $p = 1.16 \times 10^{-47}$, for the astrocytes it showed $p = 0.022$, for the neurons it showed $p = 0.644$ and for the mouse cortex it showed $p = 0.369$. and (**m**) percentage of p62 punctae and lipid droplets that show co-localization. Statistical tests between the RPE1 samples showed $p = 0.077$, for the astrocytes it showed $p = 0.004$, for the neurons it showed $p = 0.016$ and for the mouse cortex it showed $p = 0.0034$. Statistical significance was determined with a two-sample $t$-test: *=$p < 0.1$, **=$p < 0.01$, ***=$p < 0.001$. Data are presented as mean values, as the length of the box and the standard error are denoted with a whisker. On all boxes in Fig. 4j, k and m, measurements from individual cells are shown with diamonds for RPE1 cells, and measurements averaged over each individual animal are shown for the astrocytes, neurons and mouse cortex. For RPE1 cells, 81 cells were analyzed, and for RPE1 cells with ATG5KD, 150 cells were analyzed. For the wildtype astrocytes, 29 cells were analyzed for lipid droplet size and 30 cells for colocalization analysis and for the KO astrocytes, 41 cells and 40 cells, respectively, each from 3 different animals. For the wildtype neurons, 19 cells were analyzed for lipid droplet size and 55 cells for colocalization analysis from 3 different animals and for the KO neurons, 21 cells and 56 cells, respectively, from 4 different animals. For the experiments on the mouse cortex, 56 cells were analyzed for lipid droplet size and 85 cells for colocalization analysis and for the KO mouse cortex, 60 cells and 85 cells, respectively, all from 4 different animals. All experiments were performed over 3 independent biological replicates. Scale bars on overview images 10 μm, and scale bars on all zoomed images 5 μm. Source data are provided as a Source Data file.

Analysis of protein enrichment and depletion between endogeneous ATG5 and ATG5 knockdown also identified proteins related to lipid droplets or lipid and steroid metabolism: TPRG1L[64], SQLE and PIP4P1.

Finally, we inspected proteins that were possibly related to the observed calcium accumulation around the p62-lipid droplet structures. Intracellular calcium chelators, such as calmodulin or calcineurin, were neither enriched nor depleted upon ATG5 knockdown. However, we identified a calcium channel and a regulator of calcium homeostasis in the ATG5 knockdown but not in the endogeneous ATG5 sample: First, ITPR2, an ER-resident, inositol 1,4,5-trisphosphate-gated calcium channel and, second, MCUb, known as a negative regulator for the mitochondrial calcium uniporter.

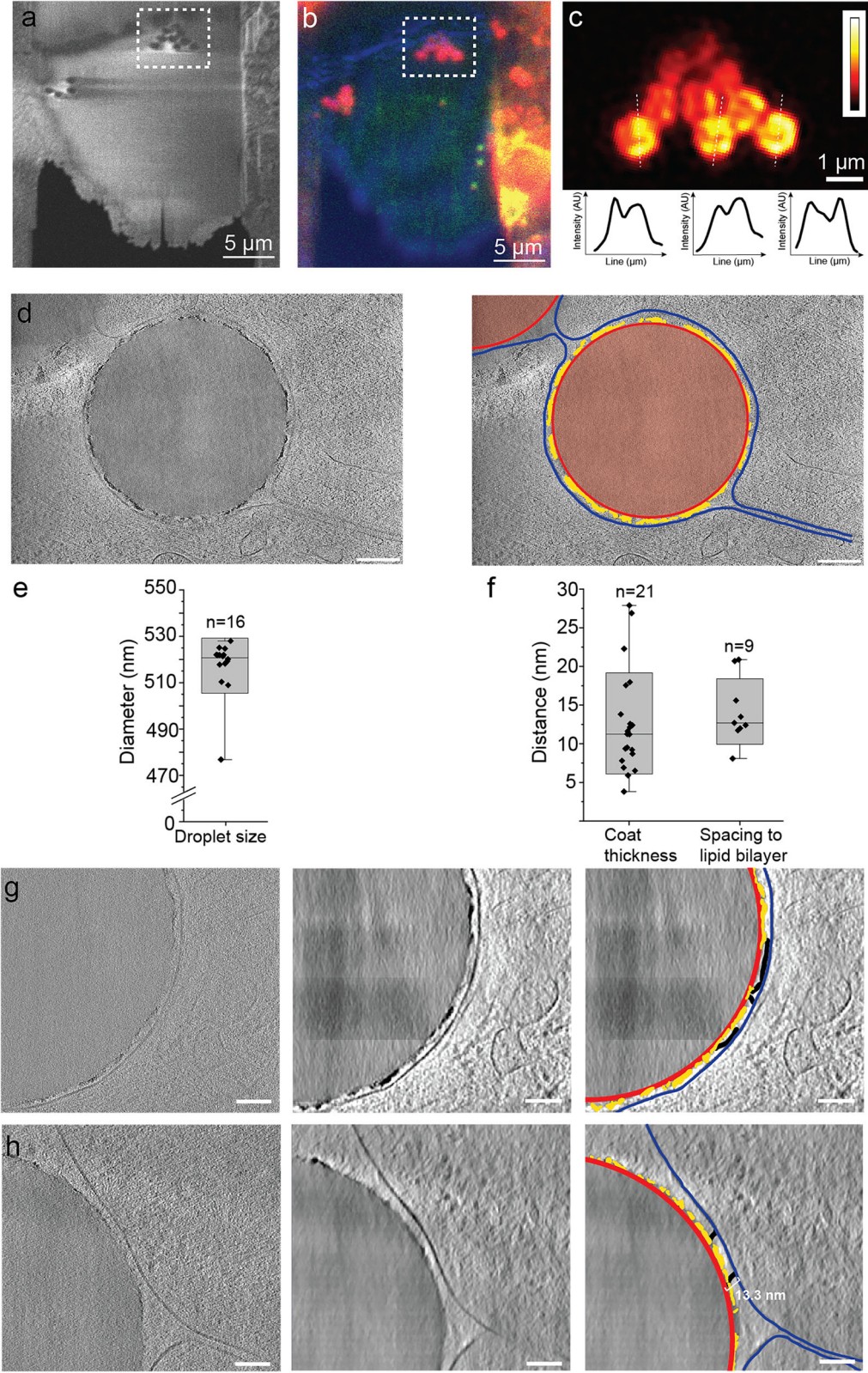

The former proteins, i.e., ITPR channels, release calcium from the ER and have been shown to affect autophagy[65]. The latter protein, i.e., MCUb, has been described to block calcium overload of nearby mitochondria, and has also been associated with the early stages of autophagy and mitophagy[66–68]. Together, our proximity proteomics experiments on the ATG5 knockdown induced p62-encapsulated lipid droplets revealed proteins such as SIRT1 and alpha-synuclein, linking lipid droplets to autophagy. Moreover, upon ATG5 knockdown, proteins involved in calcium transport from the ER, such as ITPR2 and MCUb, were found in close proximity to p62, supporting the notion that the observed calcium layer and nearby single phospholipid bilayer may represent calcium released from the ER.

**Fig. 5 | Ultrastructural in situ characterization of p62-encapsulated lipid droplets. a** Representative scanning electron microscopy image of a lamella generated from an EM grid grown with RPE1 cells. **b** Correlated fluorescence image of the corresponding lamella recorded with a reflected light channel and fluorescent channels of 595 nm for mCherry-p62 and 515 nm for autofluorescence. **c** Super-resolution cryo-confocal fluorescence image recorded at 561 nm for mCherry-p62 showed three p62-positive structures resolving a circular fluorescence signal as displayed in three corresponding intensity profiles below. The color is a heatmap, denoting signal intensity (in arb. units) (**d**) Left. Representative tomographic gray-scale z-slice through the homogeneous density sphere surrounded by a dark high-contrast layer with an enwrapped single membrane. Scale bar 100 nm. Right. Segmented and labeled structure superimposed on gray-scale z-slice, light red: lipid droplet with red: droplet surface, yellow: dark and high-contrast surrounding p62 layer and blue: single lipid membrane (right). **e** Box plot indicates a well-defined lipid droplet diameter of about 520 nm. Observed measurements are shown as diamonds, the central bar represents the median value, while the gray box corresponds to the standard deviation range. The whiskers limit the 1–99% observed data value range. The *n* value indicates the number of individual tomograms that contributed to the measurement. **f** Box plot of the measured distances of observed dark layer thickness and spacing of droplet surface to enwrapped single lipid bilayer. Observed measurements are shown as diamonds, the central bar represents the median value, while the gray box corresponds to the standard deviation range. The whiskers limit the 1–99% observed data value range. The n value indicates the number of individual tomograms that contributed to the measurement. **g** Tomographic z-slice of half a lipid droplet in raw grayscale (left), denoised grayscale (center) and superimposed labeled structures of droplet surface (red), dark layer (yellow), weaker protein density (black) and enwrapping single lipid bilayer (blue). Scale bar 50 nm. **h** Tomographic z-slice of another droplet showing the spacing of 13.3 nm between the droplet surface and a single lipid bilayer. Scale bar 50 nm. All experiments were performed on at least 7 independently prepared biological samples.

## Discussion

In the current study, we addressed the structural organization of the archetypical autophagy receptor p62/SQSMT1 at various levels of resolution. Initially, we determined the cryo-EM structure of full-length p62 at 4.5 Å global resolution revealing the double helical filament PB1 scaffold with peripheral lower-resolution densities corresponding to the ZZ-domain and C-terminal part (Fig. 1). Interestingly, the addition of LC3b led to a shortening of the filaments at higher concentrations while GST-4xUb resulted in the bundling of filaments to phase-separated entities consistent with solvent-exposed accessible binding sites to LC3b and ubiquitin through the major groove of the filament (Fig. 2). When p62 filaments were bound to LC3b-conjugated SUVs, we observed a lateral association on model membranes of the p62 filament while maintaining regular p62 oligomer arrays (Fig. 3). Moreover, we visualized p62 structures in cells using fluorescence light microscopy, correlative light and electron microscopy (CLEM) followed by cryo-ET in a p62-enriched phenotype due to ATG5 knockdown (Figs. 4, 5). The detailed ultrastructural analysis of this autophagy-stalled phenotype revealed a high-contrast layer of p62-positive structures surrounding lipid droplets due to its enriched calcium and phosphorus content (Fig. 6). Proteomic proximity labeling of the ATG5 knockdown cells revealed the upregulation of an ER-resident calcium channel and regulator supporting the notion of ER membrane in close proximity to p62 and the observed calcium layer (Fig. 7).

The here determined full-length p62 filament structure formed a double-helical filament architecture widely known from the classical DNA double helix model[69]. Our own earlier work of full-length p62 revealed the ability to form filamentous assemblies, while the cryo-EM images were acquired on a CCD camera, and single as well as double-helical filaments could not be discerned unambiguously at the limited resolution[21]. In contrast, the more recently determined PB1 (1–102) assemblies[22] acquired on direct electron detectors allowed reliable atomic model building, revealing triple or quadruple PB1 domain strands leading to a closed tube. In comparison, the here determined full-length structure revealed only two well-resolved antiparallel PB1 scaffold strands, resulting in an open major groove, while the previously determined PB1 domain-only assemblies had one or two additional PB1 strands inserted into the closed helical assembly. In the full-length structure, the space required by the additional 340 C-terminal amino acids restricts the assembly to two antiparallel PB1 strands, leaving space for a major groove. The density located C-terminally to the ZZ-domain could only be resolved at lower resolution, as it does not strictly follow the helical symmetry of the PB1 scaffold. An additional reason for the limited resolution is likely the at least partial structural disorder of the region 168–338, in agreement with previous disorder predictions[21], also reflected by the lower confidence score of the AF3 predictions included in our modeled p62 filament.

Previous studies have successfully employed bulk structural and biochemical characterization methods to characterize p62 interactions, but only for relevant p62 fragments, e.g. with the isolated PB1 domain, LIR motif peptide and UBA domain[12,70,71], but not for full-length p62 investigated here. When working with full-length p62, bulk characterization methods are often not conclusive due to the contributions of p62 self-interactions and associated heterogeneity. Therefore, we turned to single-molecule structure characterization methods such as negative staining and cryo-EM, as well as single-particle fluorescence microscopy that allow targeted identification of the relevant subpopulations for the interactions. Our single-molecule binding assays showed a shortening of the filaments upon LC3b interaction, in agreement with model predictions of the LIR motif to be accessible in solution through the major groove. In our solution studies, we also found the p62 concentrations to be critical for detecting clear binding effects such as shortening or filament bundling in the case of GST-4xUb. As a consequence of the chosen single-molecule characterization methods, for successful detection of the relevant interactions, we also had to adjust the p62 working concentrations to nM and µM for single-particle fluorescence microscopy/negative staining EM and cryo-EM, respectively. These concentration differences also explain the differences in the observed fine p62 structures after incubation with GST-4xUb, e.g., between negative staining EM and cryo-EM (see Figs. 2c, 3b). The formation of long filaments is favored at µM over the much shorter p62 fragments at nM concentration, while both morphologies are consistent with the formation of cross-linked larger p62 assemblies.

In addition, we also observed bundling of p62 filaments upon the addition of divalent cations in CaCl$_2$ and ZnCl$_2$ solutions (see Fig. 6g, h). Multivalency available in a polymer and across polymers may mediate the bundling, e.g., when several UBA domains from different filaments bind to polyubiquitin or when parts of ZZ-domains originating from multiple p62 filaments share the coordination of divalent cations such as Zn$^{2+}$ and Ca$^{2+}$. Multivalent interactions mediated by GST-4xUb have been shown to promote phase separation of p62[40]. In the cell, we expect some of these interactions to be present on membrane surfaces through membrane-linked p62 polymers and oligomers, which provide the principal benefit of high local concentrations offered for low-affinity interactions to be enhanced by high avidity binding.

In support of the low-affinity/high-avidity interaction concept, when we linked the binding partner LC3b[28] to the liposome membrane surface, upon p62 filament addition, we did not observe a complete filament disassembly; rather, we observed a lateral binding of the filament. In the cryo-EM images at physiological pH 7-8, we also found an unwinding and structural opening up of the double-helical filament architecture located on the membrane surface. In contrast, our full-length p62 filament structure was solved at pH 6 in order to obtain a

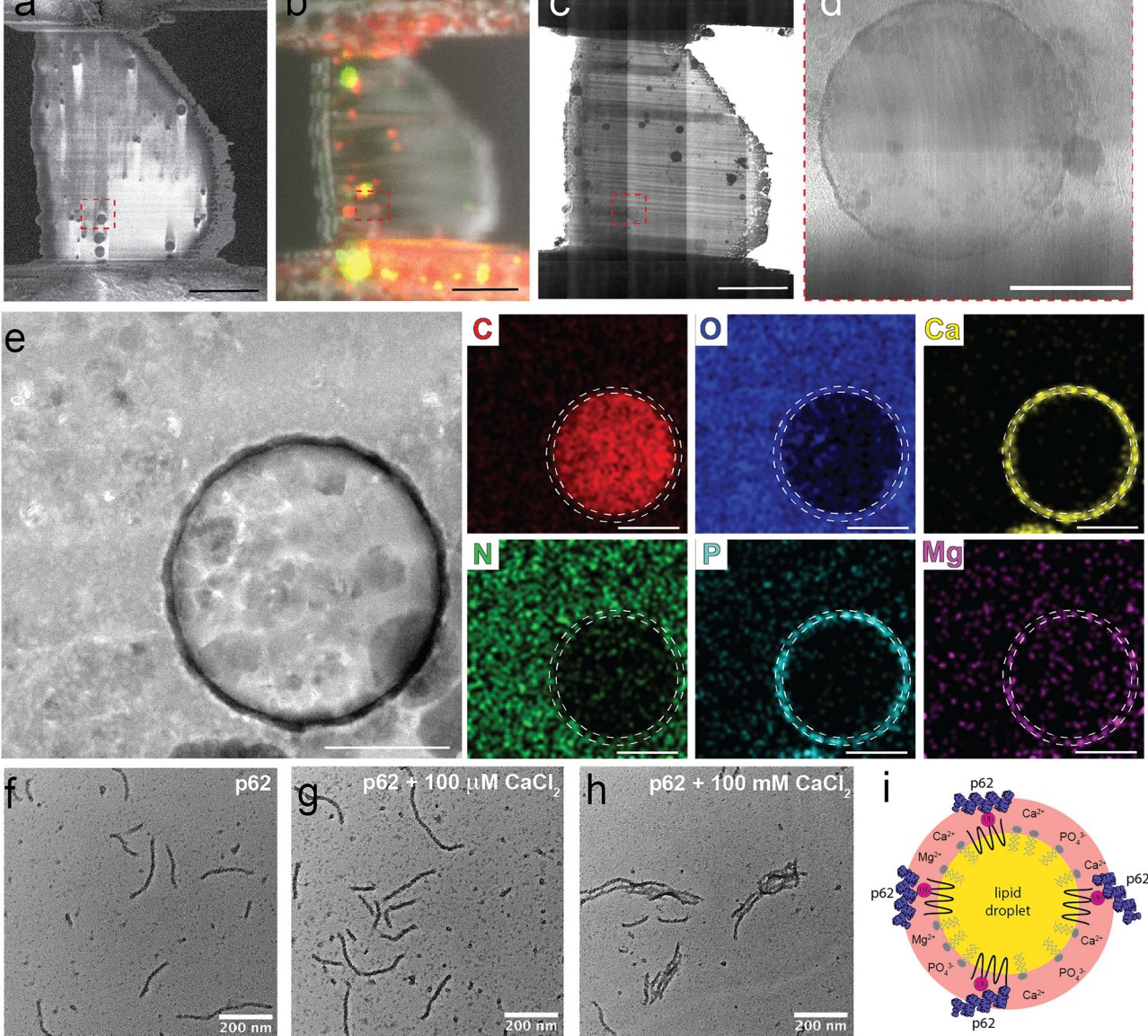

**Fig. 6 | Compositional elemental analysis of p62-encapsulated lipid droplets using energy dispersive X-ray (EDX) spectroscopy. a** Scanning electron microscopy (SEM) image of a milled lamella with an inset of characterized lipid droplet. **b** Correlated SEM image superimposed with a reflected light channel and fluorescent channels of 595 nm for mCherry-p62 and 515 nm for autofluorescence. **c** Transmission electron microscopy image of the corresponding lamella. **d** Enlarged image of p62-positive lipid droplet. **e** High-angle annular dark-field scanning transmission electron microscopy (HAADF-STEM) image of a droplet of interest with corresponding EDX signals for Carbon in red (C), Oxygen in blue (O), Calcium in yellow (Ca), Nitrogen in green (N), Phosphorus in cyan (P) and Magnesium in purple (Mg). Scale bar 500 nm. Negative stain EM images of (**f**) purified p62 filaments, (**g**) with 100 μM CaCl$_2$ and (**h**) 100 mM CaCl$_2$. **i** Model summary of analyzed p62 droplet structures: the lipid droplet devoid of any phospholipids is encapsulated by a p62-positive layer that is rich in CaPO$_4$. Experiments were done in one session, on 4 independent cellular lamellae and yielded similar results.

more homogeneous closed filament conformation following the helical symmetry. At a cytosolic pH of approximately 7.4, the p62 filaments reveal the ability to adopt an open "assembly configuration", in which previously inaccessible available binding sites in the interior will become accessible in an opened p62 "bubble" structure while still maintaining functional polymers and oligomers. At the slightly lower pH values present in the fully assembled autophagosome, estimated to be closer to pH 6, p62 will assemble in more rigid filaments. In future work, it will be interesting to characterize LC3 and poly-ubiquitin binding at various pH values to better understand this regulatory mechanism, similar to the mechanism described for arginylated substrate[45]. Therefore, the unwinding, structural opening and re-

arrangement of the filament architecture on membrane surfaces may provide a higher level of regulating access of binding partners to the multiple binding sites in p62.

After the in vitro structural and biochemical work, we turned to the human RPE1 cellular model to investigate the ultrastructural details of p62 using cryo-ET. In order to enrich p62 structures, we depleted ATG5, which is an essential member of the core machinery driving autophagy progression[1,28,53]. During autophagy, ATG5 is covalently linked to ATG12, which, together with ATG16L, is required for LC3-I conjugation to the membrane in the form of LC3-II and thereby accomplishing phagophore expansion[72]. Upon ATG5 depletion, characteristic LC3 punctae and autolysosomes cannot form, leading to an

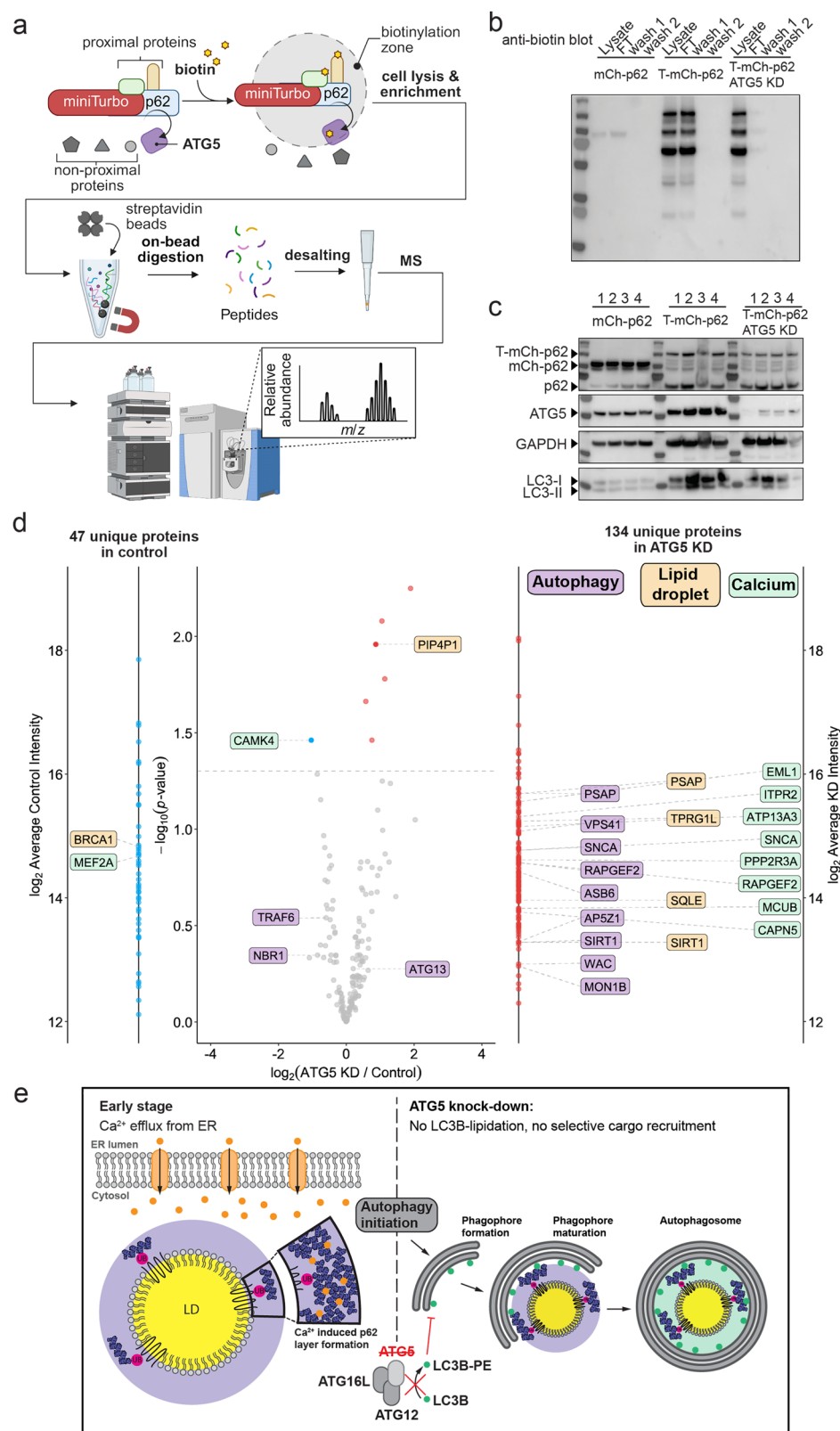

accumulation of cargo, e.g., when lipid droplet breakdown is inhibited, triacylglycerides have been shown to accumulate[54]. Moreover, in mice, ATG5 knockouts have severe consequences for neuronal development leading to delays in maturation as ATG5 appears to be essential for the survival of both adult-born hippocampal neurons[73] and cerebellar Purkinje cells[74]. We observed more and larger lipid droplets in our ATG5 knockdown cells, as well as a higher percentage of lipid droplets

co-localized with p62. We confirmed that this is not a unique phenotype limited to human RPE1 cells, as we observed the same trends in mouse ATG5 KO primary astrocytes, mouse primary neurons and mouse cortex (see Fig. 4). Based on the near-native visualization using in situ cryo-CLEM in RPE1 cells, we observed p62-positive structures wrapping around lipid droplets made of triacylglycerides giving rise to homogeneous density in the electron tomograms and characteristic

**Fig. 7 | Proximity proteomics analysis of ATG5 knockdown (KD) RPE1 cells containing p62-enwrapped lipid droplets. a** Scheme illustrating the proximity biotinylation workflow for mass spectrometry and Western blot analysis. **b** Western blots analyzing the different samples that were used in the experiments; a control sample of mCherry-p62 RPE1 cells (mCh-p62) labeled with biotin, a control sample of miniTurbo-mCherry-p62 RPE1 cells (T-mCh-p62) labeled with biotin with endogenously expressed ATG5 and ATG5 knock down + miniTurbo-mCherry-p62 RPE1 cells (T-mCh-p62 ATG5 KD) labeled with biotin. Western blot was detected with streptavidin; highlighting all biotinylated proteins in the cell lysate, flow-through and wash steps of the Streptavidin pull-down column, respectively. **c** Western blot shows p62 expression in the different cells (mCh-p62, T-mCh-p62 and T-mCh-p62 ATG5 KD, respectively), ATG5 levels, LC3 lipidation state and a GAPDH loading control. **d** Volcano plot displaying relative abundance of proteins after enrichment from ATG5 KD and control samples with endogenous amount of

ATG5. Autophagy related proteins, lipid droplet related proteins and calcium related proteins that were found significantly different abundance are highlighted. The proteins on either side of the volcano plot were only present in the control sample (in blue) or only present in the ATG5 KD sample (in red), y-axis indicates protein LFQ intensity. **e** Working model of the early events of selective autophagy captured in this study: after calcium efflux from the ER, together with calcium, p62 filaments form a coat around the lipid droplet cargo to be degraded. Due to the ATG5 KD condition preventing LC3b conjugation to phosphatidyl ethanolamine (PE) residing in the membrane, we can visualize this early structure of selective autophagy that otherwise is rapidly degraded after complete membrane enwrapment and fusion with the lysosome. All mass spectrometry experiments were performed on biological quadruplicates. The workflow figure was created in BioRender (Huesgen, lab (2025) https://BioRender.com/zh44l17).

EDX signals. Lipid droplets have been observed in the diameter ranges of 100 nm to over 1 μm in size[32], while the here observed ones had a relatively narrow distribution of 500 to 540 nm. Lipophagy has been described as being initiated by the ubiquitination of perilipin (PLIN) proteins embedded in the phospholipid monolayer surrounding the lipid droplets[75]. However, without an additional ubiquitin enrichment step, we were unable to experimentally confirm that PLIN ubiquitination recruited p62 to these lipid droplets, and we did not identify other proteins of the lipid droplet ubiquitination machinery[76] in our proteomic mass spectrometry experiments. Nevertheless, our fluorescence co-localization studies confirmed the presence of PLIN-1 in some of the p62-positive structures of the mouse cortex. LC3 and p62 were also described as forming punctae on the surface of lipid droplets[77]. In this way, they are thought to induce the formation of the autophagosomal membrane surrounding the lipid droplets. While in ATG5-depleted cells, LC3-II is not present on the membrane, p62 has been shown to still be capable of recruiting cargo, ATG9A vesicles, ATG16L vesicles and FIP200 condensates[24,78]. Possibly, the small vesicular structures we observed next to the p62 positive lipid droplets (see Supplementary Fig. 6b) may correspond to previously observed ATG9 vesicles (30–60 nm)[33,35,79,80] vesicles.

When we segmented tomography data of the ATG5-depleted cells, we observed p62 coating of various thicknesses on the lipid droplets. In close vicinity, we often found a single, unclosed membrane (see Supplementary Fig. 6a) enwrapping the coated lipid droplet. In the ATG5 knockdown condition, we do not expect a bona fide autophagy double membrane to be present. Due to the observed connections to membrane stacks, it appears plausible that we visualized recruited ER membrane unable to extend and seal off the lipid droplets due to the lack of lipidated LC3-II. Our proteomics experiments lend support to this hypothesis, as we found several ER-resident membrane proteins in close proximity to these p62-encapsulated structures. In our p62-positive structures, we also observed 13.3 nm wide protein density connecting the droplet surface/coat and the enwrapped membrane, in addition to elongated flexible density. The density is not compatible with regular helical p62 filaments used for structure determination above, as no pitch repeat distance along the long axis is discernible. Alternative assemblies, however, such as unwound or opened p62 filaments, share similar width dimensions to the measured 13.3 nm. Other potential molecules of the autophagy core machinery include ATG2A/B[81], the ULK1 complex[82] or FIP200[82], all of which have elongated shapes but are either longer or shorter than the observed densities. Clearly, more research and improved imaging features will be required to confirm the identity of the protein complexes involved in the studied p62-positive structures.

An intriguing question arises whether the here observed double-helical filaments from purified material are compatible with our cryo-ET observations in cells. Given the dense cellular environment and limited resolution of the lamellae, we could not find direct visual evidence of the helically assembled filaments. Nevertheless, we located

p62 surrounding the lipid droplet as part of a calcium-rich layer and believe it to be present as structurally modified polymeric assemblies like the ones visualized on the liposome surface or our in vitro reconstituted calcium-induced filament bundles (see Figs. 3d, 6h). As we learnt from our in vitro experiments, the tendency of p62 filament assembly/disassembly and structural properties, such as filament length, are dependent on the concentrations present in the test tube. In this context, it is an important question what the cellular concentration of p62 is. Previous reports estimated for mouse fibroblast cells and HEK293T cells the average p62 copy number to be 25,000 and 97,000, respectively[83,84]. Assuming an average cytoplasmic volume of 1500 μm[3] of an RPE1 cell[85], the resulting average concentration is between 0.1 μM/100 nM and 0.4/400 nM. This concentration is about three orders of magnitude lower than the 2 mg/ml (41.9 μM) used in this study to generate filaments for structural analysis. While the 100 nM concentrations are of the same order of magnitude as used in the nM binding studies examined by single-particle fluorescence microscopy and negative staining EM. Nevertheless, in the cell we expect significant variation in local concentrations, e.g. when p62 forms a condensate or a dense layer around a lipid droplet cargo, making local structural enrichment and effective higher concentrations possible.

The p62-lipid droplet structures we have visualized in this study were found surrounded by dark layers. We identified them to be mainly composed of calcium, phosphorus and magnesium using EDX spectroscopy. Previously, the same approach has been used on vitreous sections and other frozen hydrated samples[86–89] while in this study, we employed it on FIB-milled lamella. A dark high-contrast layer had been observed before in close proximity to lipid droplets in the context of autophagy using plastic section electron microscopy[32,54,77]. We do not claim that the previously imaged structures correspond to structures we observed; nevertheless, the resemblance is noteworthy. For some time, intracellular calcium levels have been linked to autophagy as rapamycin-induced autophagy led to an increase in cytosolic $Ca^{2+}$ concentration and a decrease in the calcium ER leak rate, both of which were independent of LC3 lipidation[90]. Moreover, the addition of a calcium chelator such as BAPTA-AM was shown to be sufficient to block autophagy initiation[68]. In a more recent study, evidence was provided that brief calcium transients are released from the ER, which trigger FIP200 punctae formation and modulate phase separation at the autophagosome assembly site[91]. In yeast, calcium has also been shown to be required for autophagy initiation under glucose starvation conditions[92]. The fact that we identified calcium transporters in our proteomics studies, such as ITPR2, further lends support to this hypothesis. ITPR2, an inositol 1,4,5-trisphosphate-gated calcium channel found in the ER membrane, has been shown to be involved in autophagy initiation and senescence[66]. Additional reports demonstrated that knockdown of its paralogue, ITPR1, led to autophagosome accumulation[93], and ITPR2/ITPR3 knockdown reduced FIP200 punctae, WIPI2 punctae and LC3 punctae under starvation conditions[91]. It

seems, therefore, plausible that in this study we directly observed calcium ions bound to the lipid droplet surface originally released from the ER by ITPR2, which help to organize the early stages of lipophagy. We were unable to identify calcium chelating proteins in our proteomics studies to retain calcium in proximity to p62 polymers. Given the demonstrated bundling capabilities of p62 itself by divalent cations in solution, it is possible that the observed dark p62-positive coat found on the cargo lipid droplet is the result of calcium transients release from the ER that orchestrate the early stages of autophagy machinery recruitment.

Given the multitude of observations in a series of biochemical and cellular experiments, we summarize our findings in a model of the early stages of selective autophagy (Fig. 7e): p62 polymers recognize the ubiquitinated surface of lipid droplets destined for degradation in proximity to the ER. In parallel, an unknown signal leads to calcium efflux from the ER that is collected by the p62 polymers surrounding the cargo with a calcium-rich layer. Additional autophagy factors are recruited, leading to the progression of autophagy, e.g. the ATG5/12/16 complex conjugating LC3b to the phosphatidyl ethanolamine (PE) membrane and thereby mediating phagophore formation and extension up to an autophagosome. In the absence of ATG5 membrane extension, LC3b is not conjugated to PE, leading to an accumulation of the calcium-rich p62-positive lipid droplet cargo structures. Further research will be required to fully establish the molecular mechanistic links between the here presented cryo-EM structure, biochemical binding properties, higher-order organization, in situ ultrastructure and early stages of selective autophagy.

## Methods

### Protein expression, purification of p62, 4x-linear-ubiquitin and LC3B

Full-length p62 (440 residues) was cloned into a pETM43 expression vector containing an N-terminal maltose-binding protein (MBP) tag and a C-terminal His-tag. Both tags featured a recognition sequence for HRV-3C protease, facilitating their removal after purification. 4x-linear-Ubiquitin, CFP-LC3B, and LC3B with an additional C-terminal cysteine residue were cloned into the pGEX expression vector containing an N-terminal Glutathione-S-Transferase (GST) tag with an HRV-3C protease cleavage site. For protein expression, E. *coli* BL21 (DE3) cells were used, with bacterial cultures grown at 37 °C in Terrific Broth (TB) medium to an $OD_{600}$ of ~2. Induction with IsoPropylThioGalactoside (IPTG) followed, and cells were left to grow overnight at 23 °C. Cells were harvested, and the pellet was stored at −80 °C until further use.

After thawing on ice, cells were resuspended in lysis buffer [50 mM HEPES, pH 8.0, 300 mM NaCl, 2.5 mM $MgSO_4$, 10 µM $ZnCl_2$, 0.5 mM TCEP and 1 tablet EDTA-free protease inhibitor cocktail] and lysed using a cell disruptor at 1.85 bar. Cell debris was removed by centrifugation, and the cleared lysate was applied to a Ni-NTA affinity column. The column was washed twice with lysis buffer containing 10 mM and 20 mM imidazole, respectively, and the protein was eluted with lysis buffer containing 300 mM imidazole. Eluted protein fractions were pooled and subsequently applied to an amylose resin that binds MBP-fusion proteins with high affinity. The column was washed with lysis buffer, and the protein was eluted with lysis buffer containing 25 mM maltose. Eluted protein fractions were pooled, and buffer was exchanged to desalting buffer [50 mM HEPES, pH 8.0, 300 mM NaCl and 0.5 mM TCEP] using Zeba desalting columns.

For the purification of LC3b and GST-4xUb, the cell pellets were resuspended in lysis buffer [50 mM HEPES, pH 7.4, 300 mM NaCl, 1 mM DTT, and 1 tablet EDTA-free protease inhibitor cocktail] and lysed using a cell disruptor at 1.9 bar. The cleared lysate was applied to Glutathione Sepharose 4 Fast Flow GST-tagged protein purification resin (Cytiva, Cat No 17513201), washed with 10 column volume lysis buffer, and eluted in lysis buffer containing 10 mM reduced glutathione. Elution fractions containing the protein of interest were pooled, further purified by size exclusion chromatography via SD200 16/600 column (Cytiva) [in 50 mM HEPES, pH 7.4, 150 mM NaCl, 1 mM DTT], and frozen at −80 °C for storage.

### p62 filament formation and labeling

Protein was concentrated to approximately 2 mg/mL using a 30 kDa Amicon concentrator and subsequently incubated with 1:2.5 mass ratio of GST-HRV-3C protease for 3 h at room temperature and overnight at 4 °C to remove the MBP and His-tag, allowing p62 filament formation. Filaments were isolated by centrifugation at 20,000 × g for 30 min and resuspended in desalting buffer. Isolated filaments were labeled with lysine residues by incubating with 1:1 molar ratio of Cy5-NHS for 1 h at room temperature and overnight at 4 °C. Excess dye was removed by centrifugation, labeled filaments were resuspended in storage buffer [50 mM HEPES, pH 8.0, 50 mM NaCl and 0.5 mM TCEP] and stored at 4 °C until further use.

### Cryo-EM structure determination of p62 filaments

Quantifoil Cu 200 mesh R1.2/R1.3 grids were glow-discharged, and 2–5 µL of the protein sample was applied in a Vitrobot Mark IV (ThermoFischer Scientific). The sample chamber was maintained at 70–100 % humidity and 4–10 °C, with a 2–6 s blotting time before plunging into an ethane/propane mixture. Data collection was performed using SerialEM[94] on a Talos Arctica 200 KeV G2 transmission electron microscope (ThermoFischer Scientific), equipped with a K3 direct electron detector camera and Quantum GIF filter (Gatan, Inc). Approximately 4000 micrographs were collected in counted super-resolution mode at 100,000x magnification with an underfocus range of −0.5 µm to −3 µm. The micrographs were motion corrected in WARP[95], and particles were picked using the deep-learning-based object detection software crYOLO[96]. Initially, the picked particle locations and micrographs were imported into RELION and SPRING[46,48], which used CTFFIND4 for CTF estimation[97]. Later, CTF estimated micrographs were imported into cryoSPARC[47] and particle locations were extracted with a 90 % overlap, yielding approximately 1,000,000 segments of a box size of 503 Å. Two rounds of 2D classification were performed in order to remove heterogeneous particles. Classes with clearly defined PB1 domains were then selected, leading to a total of 600,000 segments. The helical parameters as determined by Jakobi et al.[22] of truncated p62 helical filaments were taken as starting conditions for helical refinement in cryoSPARC, where the symmetry parameters converged (Supplementary Fig. 11). The final resolution was estimated by FSC using the 0.143 cutoff[98] and FDR-FSC criterion[99]. Maps were rendered in UCSF ChimeraX[100].

### AlphaFold3 predictions and modeling of a p62 filament

To model the p62 filament structure, we used AlphaFold3 (AF3) multimer predictions[51]: first, we tested p62 full-length sequences arranged in dimers, trimers, tetramers, up to decamers. AF3 consistently favored closed ring-like assemblies, even for small oligomers (≥3 subunits), while for larger rings the resulting p62 rings exhibited close-to-rotational symmetry. A single p62 subunit was excised from an AF3-generated p62 closed ring-like decamer with a diameter of 145 Å, approaching the diameter of the filament. To relate these predictions to the p62 filament, we fitted the PB1 domain in the corresponding well-resolved density of one strand of the filament map while including the full-length p62 model and repeated the operation for the opposite PB1 strand. Using the determined helical symmetry parameters of 11.0 Å helical rise and a −26.1° helical rotation, we propagated the two subunits along the helical axis, generating a double-helical filament model composed of repeating p62 units. This helical assembly yielded a plausible model of the p62 filament, with subunits well fitting the experimental cryo-EM density while maintaining interaction geometries close to the AF3-predicted oligomers.

## Single-particle fluorescence imaging in TIRF mode

Surface imaging was performed with an inverted microscope (Olympus IX-81, Olympus Corp., Shinjuku, Japan) in total internal reflection fluorescence (TIRF) mode. Molecules were excited using lasers at 488 and 639 nm (Sapphire 488–200 and Obis 637–140, Coherent Inc.), focused off-axis into the back-focal plane of an oil-immersion objective (Olympus UApoN, 100x, NA = 1.49, Olympus Corp). The excitation power and corresponding intensity at the sample stage were 3.25 mW and 10.2 W/cm² for the blue laser, and 2.2 mW and 6.8 W/cm² for the red laser. The imaging was performed on an EMCCD camera (Andor iXon Ultra 888, Andor Technology) with 50, 100, 40 and 100 ms exposure time for Supplementary Videos 1–4, corresponding to 20, 10, 25 and 10 fps. Midway through acquiring Supplementary Video 2, the illumination was switched from blue to red laser to observe the same p62 filament with LC3b bound to it. Each video consists of 200 frames. Image analysis and video processing were carried out using Fiji (ImageJ, version 2.14.0)[101]. The emitted fluorescence was collected with the same objective (objective-based TIRF) and passed through a dual-band dichroic mirror as well as a dual-band emission filter (ZT488/640rpc and ZET488/640 m, Chroma Technology Corp). p62 filaments-Cy5 and CFP-LC3b were diluted to concentrations of a few nM, and mixed in a 1:1 molar ratio. GST-4xUb was added at a concentration of 5 µM.

## Negative staining electron microscopy

For negative staining, 3.5 µL of the sample was applied onto glow-discharged carbon-formvar-coated 300-mesh copper grids and allowed to absorb for 2 min before being blotted according to the side-blotting method. The sample was subsequently stained for 30 s with 6 µL 2% uranyl acetate, blotted and allowed to dry. Negatively stained samples were examined on a Talos L120C G2 transmission electron microscope (ThermoFisher Scientific), operated at 120 kV. Micrographs at high magnification (45,000x and 92,000x) were collected on a 4k x 4k Ceta 16 M CEMOS camera using TEM Imaging & Analysis Software (TIA, ThermoFisher Scientific).

## Statistical analysis of filament length distribution

We used Origin2021b to perform the Kolmogorov-Smirnov (K-S) test for the two datasets of filament lengths (p62 filaments without and with LC3b present). We sorted the two datasets in ascending filament length order and computed the cumulative distribution function (CDF) for both datasets. CDF increases from 0 to 1 from the smallest to the largest value in the distribution. The CDFs from both datasets were overlaid in a plot and the largest absolute difference between them was computed (Supplementary Fig. 12).

$$D = \max |F_1(x) - F_2(x)| \tag{1}$$

where $F_1(x)$ and $F_2(x)$ are the CDFs of each dataset.

Then we computed the K−S test statistic K

$$K = D \times \sqrt{(n_1 n_2)/(n_1 + n_2)} \tag{2}$$

where $n_1$ and $n_2$ are the sizes of each dataset.

The approximate p-value was calculated as

$$p \approx e^{-2K^2} \tag{3}$$

## Cryo-electron tomography of reconstituted p62 filaments

p62 filament cryo-EM grids were prepared as described above. Tilt series were recorded on a 200 kV Talos Arctica G2 transmission electron microscope (ThermoFisher Scientific), equipped with a Bioquantum GIF (Gatan, Inc), a Volta Phase Plate and a K3 direct electron detector (Gatan, Inc) using SerialEM (Mastronarde, 2005). Each tilt was recorded as a movie at 63,000x magnification with a 0.681 Å/pixel size

in super resolution mode, corresponding to a dose of approximately 2.0 e/Å² at 0° tilt, 10 frames per tilt, and a total dose per tilt series of approximately 130 e/Å². All tilt series were recorded from −54° to 54° with 3° increments, using a dose-symmetric bidirectional acquisition scheme including 'the varied intensity' option (i.e. higher tilt angles received higher doses, as 1/cosine of the tilt angle) in SerialEM and a nominal defocus of −1.2 µm. The raw data were motion corrected and CTF corrected using WARP (version 1.0.9)[95] and the tilt series were reconstructed using IMOD[102].

GST-4x-linear-Ubiquitin and p62 filaments were prepared as described above. 15 µM of GST-4x-linear-Ubiquitin were mixed with 1 µM p62 filaments. After 4 min incubation time at 4 °C, the formed phase separation droplets were negatively stained as described above or plunge frozen with a Vitrobot Mark IV (ThermoFisher Scientific). 50 nm SUVs were prepared containing 5% DSPE-PEG(2000) maleimide (Avanti Polar lipids, Cat No 880126) in DOPS (Avantiv Polar lipids, Cat No 840035). The liposomes were incubated with 100 µM LC3b-Cys for 16 h at 4 °C before removing unbound LC3b-Cys by size exclusion chromatography on Sephacryl S-500 at Äkta Micro (Cytiva). A total of 2 mg/ml of liposomes was mixed with 1 µM p62 filaments and incubated for 5 min at 4 °C. The sample was plunge frozen using a Leica GM2 plunge freezer and imaged in a 300 kV Titan Krios G4 equipped with a BioContinuum energy filter and K3 camera at 64,000x using Tomo5 software (ThermoFisher Scientific). The tilt series were recorded from −60° to 60° with a 3° increment in a dose-symmetric bidirectional acquisition scheme. An electron dose of 4.0 e⁻/Å²/tilt (at 0° tilt) with 6 frames per tilt resulted in a total dose per tilt series of approximately 160 e⁻/Å². The defocus was varied between −2.0 to −4.0 µm. The images were motion corrected in WARP[95] and tomograms reconstructed with AreTomo[103]. Overall binning was 16x and resulted in a pixel size of 11.2 Å. Tomograms were denoised and missing wedge corrected with IsoNet[104].

## Cell culture and transfection

Cell culture experiments were carried out in human RPE1 cells (hTERT RPE1, CRL-4000, ATCC) cultured at 37 °C and 5% CO₂ in DMEM GlutaMAX media (Gibco, Cat No 31331028) supplemented with 10% FCS (Sigma-Aldrich, Cat No F7524) and Pen/Strep (Gibco, Cat No 15140122). For confocal microscopy experiments prior to plunge-freezing, the culturing media were substituted with media of low fluorescence background: Fluorobrite DMEM (Gibco, Cat No A1896701). Additional experiments were performed on RPE1 cells stably expressing human mCherry-p62. Transient expression of mCherry-p62 was executed by transfection with FugeneHD (Promega, Cat No E2311) and a modified pDEST vector containing the human WT p62 gene fused with an N-terminal mCherry sequence (Pankiv et al., 2007). Knockdown experiments were carried out by transfection with RNAiMAX (ThermoFisher, Cat No 13778075) and ON-TARGETplus SMARTpool siRNA against ATG5 (Dharmcon Horizon Discovery, Cat No L-004374-00-0005), GAPDH (Dharmcon Horizon Discovery, Cat No D-001830-10-05) and a non-targeting control (Dharmcon Horizon Discovery, Cat No D-001810-10-05). Follow-up experiments were performed 72 h after a reverse transfection with 75 nm siRNA. Lipid droplets were stained with 1:500 Lipi-Blue (Dojindo, Cat No LD01-10) for 24 h. Lysosomes were stained with the CytoPainter Lysosomal staining kit (Abcam, Cat No ab112136).

## Preparation of primary postnatal cortical culture and Immunocytochemistry

All animal experiments were approved and performed according to the regulations of the LANUV, NRW, Germany (AZ: 81-02.04.2020.A418, 81-02.05.40.20.075, 81-02.04.2021.A132, Anzeige §4.20.021) guidelines. Mice were maintained in a pathogen-free environment in ventilated polycarbonate cages and housed in groups of five animals per cage with constant temperature and humidity at 12-h/12-h light–dark cycles. Food

and water were provided ad libitum. *Atg5*flox:*Slc32a1*-Cre:tdTomato (ATG5 cKO) mice were described previously[56,74]. The following genotyping primers were used to genotype the animals: ATG5 forward 1 (GAA TAT GAA GCC ACA CCC CTG AAA TG), ATG5 forward 2 (ACA ACG TCG AGC ACG CTG GCG AAG G), ATG5 reverse (GTA CTG CAT AAT GGT TTA ACT CTT GC), Cre_1(GAA CCT GAT GGA CAT GTT CAG G), Cre_2(AGT GCG TTC GAA CGC TAG AGC CTG T), Cre_3(TTA CGT CCA TCG TGG ACA) and Cre_4(TGG GCT GGG TGT TAG CC). In accordance with the SAGER guidelines, we have ensured that both male and female mice were included in all experimental groups. Cortical and hippocampal neurons from postnatal pups (P1–5) were isolated as previously described[55] and plated on PDL-coated coverslips or dishes. Neurons were grown in a MEM (Gibco) medium, containing 0.5% Glucose (Sigma), 0.02% NaHCO3 (Roth) and 0.01% Transferrin (Merck), which was further supplemented with 5% FBS (Sigma), 0.25% L-GlutMax (100×, Gibco), 2% B27 (50×, Gibco), 1% Pen/Strep (10000 Units/ml Penicillin, 10,000 μg/ml Streptomycin, Gibco). After two days in vitro 2 mM AraC was added to the culture medium to limit glial proliferation. To initiate homologous recombination, primary neurons isolated from floxed animals expressing a tamoxifen-inducible Cre recombinase were treated with 0.4 μM (Z)−4-hydroxytamoxifen (Sigma) immediately after plating. After 24 and 48 h, cells were treated with 0.2 μM and 0.4 μM of tamoxifen, respectively, during medium renewal. Ethanol was added to control neurons (WT) in an amount equal to the tamoxifen. The DIV 14–16 postnatal neurons were fixed in 4% PFA/sucrose in phosphate-buffered saline (PBS) for 15 min at RT. Cells are then washed 3x with PBS and then blocked in 5% NGS in 0.3% Triton X-100 for 1 h at RT. Primary antibodies diluted in 5% NGS and 0.3% Triton X-100. Coverslips were rinsed 3x PBS and incubated with corresponding secondary antibodies diluted in 5% NGS and 0.3% Triton X-100, and incubated for 30 min at RT. The following antibodies were used: polyclonal guinea pig anti-p62 (1:1000, Progen, cat no GP62-C), Goat anti-Guinea Pig Alexa Fluor 568 (1:1000, Invitrogen, cat no A-11075), and Donkey anti-Rabbit Alexa Fluor 647 (1:1000, Invitrogen, cat no A-31573). Cells were washed 3x in PBS and then incubated in 1:1000 Lipidspot488 (Biotium, cat no #70065-T) for 15 min at RT. Following incubation, cells were directly mounted on glass coverslips. Post-natal neurons were imaged using with Leica SP8 confocal microscope (Leica Microsystems) equipped with a 63 × /1.32 oil DIC objective and a pulsed excitation white light laser (WLL; ~ 80-ps pulse width, 80-MHz repetition rate; NKT Photonics) at a resolution of 1024 × 1024 pixels in sequential scanning frame-by-frame mode. A single Z plane was yielded. For quantitative analysis of fluorescent puncta, each channel of interest was processed by subtracting background and the area of the neuron and/or cell body was manually selected using ImageJ (Version 1.54p) selection tools (ROI). Fluorescent puncta were counted with the ImageJ Cell Counter Plugin. Fluorescent puncta size of the lipid droplets (Lipidspot488) was determined by applying the autothreshold "Huang" algorithm in ImageJ (Version 1.54p) and analyzed using the "Analyze particles" ImageJ (Version 1.54p) plugin to get average puncta size per cell in μM².

## Immunohistochemistry on perfused brain sections

All animal experiments were approved and performed according to the regulations of the LANUV, NRW, Germany (AZ: 81-02.04.2020.A418, 81-02.05.40.20.075, 81-02.04.2021.A132, Anzeige §4.20.021) guidelines. Mice were maintained in a pathogen-free environment in ventilated polycarbonate cages and housed in groups of five animals per cage with constant temperature and humidity at 12-h/12-h light–dark cycles. Food and water were provided ad libitum. *Atg5*flox:*Slc32a1*-Cre:tdTomato (ATG5 cKO) mice were described previously[56,74]. The following genotyping primers were used to genotype the animals: ATG5 forward 1 (GAA TAT GAA GCC ACA CCC CTG AAA TG), ATG5 forward 2 (ACA ACG TCG AGC ACG CTG GCG AAG G), ATG5 reverse (GTA CTG CAT AAT GGT TTA ACT CTT GC), Cre_1 (GAA CCT GAT GGA CAT GTT CAG G), Cre_2(AGT GCG TTC GAA CGC

TAG AGC CTG T), Cre_3(TTA CGT CCA TCG TGG ACA) and Cre_4(TGG GCT GGG TGT TAG CC). In accordance with the SAGER guidelines, we have ensured that both male and female mice were included in all experimental groups. Mice at 12–15-week were euthanized by an overdose of 1.2% ketamine, and 0.16% xylazine in PBS (i.p., 10 μl per 10 g body weight) and transcardial perfusion was performed as previously described[105]. Brains were carefully dissected and postfixed in 4% PFA overnight before being processed for immunohistochemistry as previously described[105]. Corresponding horizontal 40 μm sections from WT and KO littermates were washed 3x in PBS for 5 min each. Sections were blocked with 10% NGS/NDS in 0.5% PBS-T for 1 h at RT. Primary antibodies were incubated in sections in 3% NGS/NDS and 0.3% PBS-T overnight at 4 °C. Sections were washed 3 × 10 min in 0.3% PBS-T before incubation in fluorescence-labeled secondary antibodies in 3% NGS/NDS and 0.3% PBS-T for 1 h at RT, protected from light. The following antibodies were used: polyclonal guinea pig anti-p62 (1:1000, Progen, cat no GP62-C), recombinant rabbit anti-perlipin1 (1:500, Proteintech, cat no 83905-4-RR), Donkey anti-Rabbit Alexa Fluor 647 (1:1000, Invitrogen, cat no A-31573) and Goat anti-Guinea Pig Alexa Fluor 488 (1:1000, Invitrogen, cat no A-11073). Sections were imaged at a Leica SP8 confocal microscope (Leica Microsystems) equipped with a 63 × /1.32 oil DIC objective and a pulsed excitation white light laser (WLL; ~ 80-ps pulse width, 80-MHz repetition rate; NKT Photonics) at a resolution of 1024 × 1024 pixels in sequential scanning frame-by-frame mode. Stacks of 30 optical sections were acquired, with a fixed stack size of 0.25 μm. For quantitative analysis of fluorescent puncta, each channel of interest was processed by subtracting background and the area of the neuron and/or cell body was manually selected using ImageJ (Version 1.54p) selection tools (ROI). Fluorescent puncta were counted with the ImageJ Cell Counter Plugin. Fluorescent puncta size of the lipid droplets (Perlipin1) was determined by applying the autothreshold "Huang" algorithm in ImageJ (Version 1.54p) and analyzed using the "Analyze particles" ImageJ (Version 1.54p) plugin to get the average puncta size per cell in μM².

## Western Blot

Cells were lysed by a quick wash with ice-cold PBS (Gibco, Cat No 10010056) and a subsequent 15 min incubation in ice-cold RIPA buffer (Sigma-Aldrich, Cat No R0278-50mL) supplemented with a protease inhibitor cocktail (Roche, Cat No 11836170001). Cell lysate was collected with a cell scraper before removing the cell debris by centrifugation. Total protein content was quantified and corrected using a Pierce 660 nm assay (ThermoFisher, Cat No 22662) before boiling the sampling in a reducing sample buffer and separation on a 4–15 % mini-PROTEAN Precast SDS-PAGE (Bio-Rad, Cat No 4561085). Transfer to PVDF membrane was performed using a Trans-Blot pack (Bio-Rad, Cat No 1704156) and the membrane was blocked for 2 h in 2% BSA in TBST at 4 °C. The membrane was incubated overnight with primary antibody in 1 % BSA in TBST at 4 °C, washed 3 times in TBST before the secondary antibody was added for 2 h in 1% BSA in TBST at 4 °C. Prior to imaging on the Azure Sapphire RGB Biomolecular Imager (Azure Biosystems), the membrane was washed 3 times in TBST and incubated in Azure Radiance Q HRP (Biozym, Cat No 512101). The following antibodies were used: polyclonal rabbit anti-p62 (MBL International, Cat No PM045), monoclonal rabbit anti-ATG5 (Abcam, Cat No ab108327), monoclonal mouse anti-GAPDH (Invitrogen, Cat No MA1-16757), polyclonal rabbit anti-LC3 (Abcam, Cat No ab48394), goat anti-mouse HRP conjugate (ThermoFisher, Cat No 32230), and goat anti-rabbit HRP conjugate (ThermoFisher, Cat No 31460). Band intensity quantifications were performed using ImageLab (BioRad, version 6.1.0).

## Cellular tomography grid preparation

RPE1 cells were seeded on micropatterned grids before plunge freezing using recently described protocols[106,107]. Briefly, 200 mesh gold Quantifoil R2/1 grids with a SiO₂ support layer (Quantifoil, Cat No

N1-S15nAu20-01) were glow discharged and treated with PLL-g-PEG (SuSoS, PLL(20)-g[3.5]- PEG(5)) overnight. Next, the grids were washed in PBS and placed in drops of 14.5 mg/mL PLPP (Alvéole, Cat No PLPP), and 20 µm circles in the center of 10 × 10 grid squares were micro-patterned with a PRIMO photopatterning device (Alvéole, Photo-patterning Device consisting of a DMD + laser 375 nm, 75 mW module, and driving electronics). The grids were then washed thoroughly in PBS and incubated in human fibronectin immediately before use (Gibco, Cat No PHE0023). The cells were trypsinized (Gibco, Cat No 25300054), passed through a 40 µm cell strainer (PluriSelect, Cat No 43-10040-40) and counted before seeding ~200,000 cells in a 30 mm glass-bottom dish (Greiner Bio-One, Cat No 627860) containing the micropatterned grids in a small amount of DMEM. The cells were allowed to attach to the grids for 4–5 h before vitrification in liquid ethane using a Vitrobot Mark IV (ThermoFisher Scientific). The following settings were used; a chamber temperature of 37 °C, a chamber humidity of 85% humidity, backside blotting with an 8.5 s blot time and blot force of −10 and a drain time of 1 s.

### Confocal microscopy
RPE1 cell phenotypes were studied using a Zeiss LSM710 laser scanning confocal microscope (Carl Zeiss GmbH). The cells were grown in 30 mm glass-bottom dishes with 4 compartments (Greiner Bio-One, Cat No 627870) and imaged at 37 °C using a Zeiss 63× oil immersion objective (Plan-Apochromat, DIC, 1.4 NA). Lipi-Blue was excited with a diode laser at 405 nm, and fluorescent emission was detected after passing a suitable emission filter (410–500 nm) using a Multi Alkali photomultiplier tube. The lysosomal stain was excited using an argon ion laser at 488 nm, and emission was detected using an emission filter (499–541 nm). MCherry-p62 was detected using a 543 HeNe543 laser and an emission filter set to 579–696 nm using a GaAsP photomultiplier tube. Image analysis and quantification were done using Fiji (ImageJ, version 2.14.0)[101] and a self-written routine in CellProfiler (version 4.2.6)[108]. Measurements were exported as.csv files and analyzed in OriginPro 2023 (OriginLab Corporation, version 10.0.0.154). Vitrified grids and thinned lamellae were studied using a Zeiss LSM800 upright laser scanning confocal microscope with a cryostage (Linkam CMS196). Super-resolution images were acquired using a Zeiss 100x air immersion objective (EC Epiplan NeoFluar, DIC, NA = 0.75) and an AiryScan Detector. MCherry-p62 was detected using a 561 nm diode. Image processing was done using Zen Blue (Carl Zeiss GmbH, version 3.3.89).

### Correlative light and electron microscopy
Cryo-lamellae were made in a correlative manner using the in-chamber METEOR fluorescence microscope (Delmic, Cat No 2707-999-0014-2) equipped with a LMPLFLN 50×, NA = 0.5, WD = 10.6 mm objective lens in the Aquilos2 cryo-FIB-SEM system (ThermoFisher Scientific). In short, micropatterned grids with RPE1 cells were loaded under cryo-genic conditions, taking special care to avoid contamination build-up from air humidity. Cells were located using the electron beam, and the sample was placed at eucentric height using MAPS software (ThermoFisher Scientific, version 3.19). Cell phenotype was confirmed by imaging each cell in the METEOR fluorescence microscope. Each cell was imaged using the focused ion beam, and the location of the final 150 nm lamella was marked in AutoTEM (ThermoFisher Scientific, version 2.4.1). The lamellae were generated using a step-wise process; the cell was thinned to 1 µm with a 0.5 nA milling current, then thinned to 600 nm with a 0.3 nA current, next to 300 nm with a 0.1 nA current and finally thinned to 200 nm with a 50 pA ion beam. The final polishing step, in which the lamella was thinned to 120-180 nm, took place with a 30 pA ion beam. The final lamella was once again imaged using the METEOR fluorescence microscope, and these images were used to determine where tomograms would be recorded. The lamella was coated with a thin platinum layer; sputtered for 7 s at 7 mA, to prevent sample charging as previously described (Berkamp et al., 2023). Next,

the samples were transferred under cryo-conditions to a 200 kV Talos Arctica G2 transmission electron microscope (ThermoFisher Scientific), equipped with a Bioquantum GIF (Gatan, Inc) and a K3 direct electron detector (Gatan, Inc). Tilt series were recorded on p62 positive areas of the lamellae, using SerialEM (version 4.2.0)[94] as movies at 24,000x magnification with a 1.7435 Å/pixel size in super resolution mode and a dose of approximately 2.0 e/Å² at 0° tilt, 10 frames per tilt, and a total dose per tilt series of approximately 130 e/Å². All tilt series were recorded from −54° to 54° with 3° increments starting at the pre-tilt of the lamella, using a dose-symmetric bidirectional acquisition scheme with weighted dose (i.e. higher tilt angles received higher doses, as 1/cosine of the tilt angle), tilt angle and a nominal defocus of −7 µm. The raw data were motion corrected and CTF corrected using WARP (version 1.0.9)[95], and the tilt series were reconstructed using Aretomo2 (SBGrid)[103,109]. To aid the cryo tomogram visualization, they were deconvolved, denoised, and the missing wedge compensated using IsoNet[104] and cryoCARE[110]. To aid in segmentation a pre-trained U-Net model for non-denoised data in MemBrain-Seg (version 9b)[111] was used and the processed tomograms were then loaded into Amira 3D (ThermoFisher Scientific, version 2022.2) for semi-manual segmentation. Tomograms were visualized using the IMOD software package (version 4.11.20)[102]. To aid in targeting the EDX spectra, high dose, single tilt micrographs were recorded on p62 positive areas of the lamellae at 24,000x and a dose of approximately 40 e⁻/Å²/micrograph and 180 frames per tilt, with each a dose of approximately 0.24 e⁻/Å². Motion correction for these micrographs was done in Relion (SBGrid, version 4.0.1)[112]

### Energy-dispersive X-ray (EDX) spectroscopy
The in situ EDX experiments were carried out on a probe-corrected 200 kV FEI Titan microscope (ThermoFisher Scientific) fitted with a Super-X G2 quadrant detector EDX system (Bruker) and a Simple Origin dual grid cryo transfer holder (Simple Origin, Inc). The microscope was operated under cryo-conditions in STEM mode with a 1.5 mrad convergence angle, a 1 nm probe size at a magnification of 55,000, resulting in a 2.9 nm pixel size. EDX spectra were recorded with a 50 µs dwell time and 50 frames, resulting in a total dose of 500 e/Å², a camera length of 3.5 m and a detector dead time of approximately 70%. We found 500 e⁻/Å² to be a good trade-off between a high signal-to-noise ratio in the EDX spectra while still maintaining the ultrastructural integrity of the sample. EDX spectra processing was done in Velox (ThermoFisher Scientific, version 3.14). Atomic map representations were rendered with a 12 px pre-filter average and a 12 px post-filter average for the p62 structures and a 25 px pre-filter average and a 9 px post-filter average for the lipid droplet and lysosome control experiments. The spectra were generated by segmenting the p62 coat in the HAADF-STEM micrograph and segmenting the same number of pixels in the lipid droplet core and a random area of the cytosol. This same process was followed for the control experiments.

### Proximity proteomics
Lentivirus was ordered with a vector corresponding to V5-miniTurbo-mCherry-p62-IRES-puro (Sirion Biotech GmbH). The virus was added to WT RPE1 cells, clones were puromycin selected, and transduction confirmed via fluorescence microscopy. The cells were seeded in T75 flasks, and ATG5 was knocked down using siRNAs (ON-TARGETplus SMARTpool siRNA against ATG5, Dharmacon Horizon Discovery), and the cells were incubated for 72 h. Next, 500 µM biotin was added (Sigma-Aldrich, Cat No B4501-1G,) and 5 mM ATP (AppliChem, Cat No A1348,0100) and the samples were incubated for 2 h at 37 °C. The cells were placed one ice, washed in ice-cold PBS (Gibco, Cat No 10010056) and harvested with a cell scraper and RIPA buffer (Sigma-Aldrich, Cat No R0278-50mL) supplemented with protease inhibitor cocktail (Roche, Cat No 11836170001) and 1 mM PMSF. Cell debris was removed

via centrifugation. All experiments were done in quadruplicates, on mCherry-p62 RPE1 cells, miniTurbo-mCherry-p62 RPE1 cells and ATG5 KD + miniTurbo-mCherry-p62 RPE1 cells. The lysates were incubated with 200 μL washed Pierce Streptavidin Magnetic Beads (Thermo Fisher Scientific, Cat No 88817) overnight at 4 °C. The beads were then washed with 3 × 500 μL 100 mM Hepes, pH = 7.4, 1 mL 1 M KCl, 1 mL 0.1 M freshly made $Na_2CO_3$, 3 × 1 mL 3 M urea and finally the beads were suspended in 100 μL 3 M urea in 10 mM Tris, pH = 8. DTT was added to the final concentration of 10 mM, and the sample was incubated for 30 min at 56 °C with 1000 rpm shaking. Next, the sample was alkylated with 50 mM CAA (Sigma-Aldrich, Cat No 22790-250 G) for 30 min in the dark. The reaction was quenched with 50 mM DTT for 20 min, and the beads washed twice with 100 uL 2 M urea. On-bead digestion was performed overnight at 37 °C in 50 μL 100 mM HEPES pH = 7.4 using a 1:100 trypsin to proteome ratio (Promega, Cat no V5111). The supernatant was collected and pooled with 2 × 100 μL 2 M urea wash steps. The samples were acidified with 2% TFA (Sigma Aldrich, Cat no 1082180050), to pH < 3. StageTips (Affinisep, Cat No Tips-RPS-M.T1.200.96) were equilibrated as follows: they were washed with pure methanol, with 0.1% formic acid in 80% acetonitrile and then twice with 0.1% formic acid. Each sample was split over 6 StageTips and centrifuged at 500 × $g$ for 2 min, and then washed twice in 30 uL 0.1% formic acid and centrifuged at 500 × $g$ for 2 min. Peptides were eluted with 60 μL of 60% ACN, 5% NH3 and vacuum-concentrated with the concentrator plus (Eppendorf). The samples were stored at −20 °C, and right before mass spectrometry measurement, reconstituted in 0.1% TFA. The concentration was determined with a Nanodrop before injecting the peptides on a Q Exactive Orbitrap (Thermo Fisher Scientific) coupled to an UltiMate® 3000 UHPLC-System (Thermo Fisher Scientific) ran in positive mode utilizing a μPACTM C18 pillar array column (200 cm bed length, Thermo Fisher Scientific) heated to 40 °C. The 140 min gradient was applied in the following solvent B concentration: 6% to 5 min, 6–10% (5–10 min), 10–45% until 110 min, 45–99% until 118 min, 99% for 7.9 min, 99–6% in 0.1 min, and 6% for 14 min. Mobile phase A consisted of 0.1% formic acid (LC-MS grade) in water, and mobile phase B consisted of 0.1% formic acid in 86% acetonitrile (LC-MS grade). After peptide separation, electrospray ionization (ESI) was used as the ionization method. Therefore, a Nanospray Flex ion source (Thermo Fisher Scientific), liquid Junction™ PST-HV-NFU (MS Wil) and a fused silica emitter (20 μm inner diameter, 360 μm outer diameter, MicroOmics Technologies LLC) were used. A spray voltage of 1.8 kV and a capillary temperature of 250 °C were applied. Full or overview MS scans were acquired in the m/z range of 385–1050 with a resolution of 70000 (at m/z 200) and an automatic gain control (AGC) target of $3 × 10^6$ ions with a maximum injection time of 60 ms. Each DIA cycle consisted of one full MS scan followed by 50 DIA windows with 12.5 m/z isolation width overlapping by 0.25 m/z covering the mass range of 400-1000. Each DIA window had a resolution of 35000, an AGC target of $1 × 10^6$ ions, and a maximum injection time of 60 ms. The peptides were fragmented by higher-energy collision-induced dissociation (HCD) with a normalized collision energy (NCE) of 26. Raw data was first converted by MSConvertGUI (64-bit, version 3.0.21229-9668f52) to the mzML format files using PeakPicking filtering. FragPipe (version 22.0) was used for database search by matching DIA spectra to the FASTA file path according to Proteome (i.e. UniProt H. sapiens canonical and isoform protein sequences, downloaded in May 2023), added with decoys and contaminants. MSFragger settings included peptide digestion with trypsin; for a peptide length from 7 to 45, a number of missed cleavages was set to 2, a precursor mass range from 300 to 5000 and the mass tolerance for precursor and fragment ions was set to 20 ppm. Commonly, methionine oxidation (M) and N-terminal acetylation ([^) were set as variable modifications and cysteine alkylation as a fixed modification. Validation settings of DIA_speclib_Quant were used, including MSBooster for deep learning prediction Peptide Spectrum Match (PSM) Validation with Percolator

to increase confident identifications, and Protein Interference with ProteinProphet to generate the protein list with a 1% false discovery rate (FDR). The report.pg_matrix.tsv file from the FragPipe output was processed with RStudio (R version 4.4.0 (2024-04-24) x 86_64-apple-darwin20) using the packages limma and dplyr. A $log_2$ transformation was applied to the data to stabilize variance and normalize the data distribution. Next, a threshold was defined to filter rows based on the presence of valid (non-missing) values. Only rows where at least 75% of the values in each specified condition are valid were retained. The linear model was fitted to the data using the lmFit function from the limma package. An empirical Bayes moderation was performed to obtain more stable estimates of the variance. Moderated $t$-statistics and $p$-values for the comparison between two conditions were extracted, and the $p$-values were also adjusted for multiple hypothesis testing using the FDR method[113]. Finally, the log fold change between the two conditions was calculated for each comparison. All proteins found in control samples from cells expressing mCherry-p62 (mCh-p62) were considered non-specific contaminants and removed from the dataset before comparison of protein abundance in cells expressing T-mCh-p62.

## Ethical statement
All animal experiments were approved and performed according to the regulations of the LANUV, NRW, Germany and ethics committee guidelines (AZ: 81-02.04.2020.A418, 81-02.04.2023.VG076, 81-02.04.2021.A132, Anzeige §4.24.030).

## Reporting summary
Further information on research design is available in the Nature Portfolio Reporting Summary linked to this article.

## Data availability
The EMDB accession number for the p62 cryo-EM map is EMD-52134, and the corresponding PDB-ID 9HGE for the fitted PB1 coordinates. The mass spectrometry proteomics data have been deposited to the ProteomeXchange Consortium via the PRIDE [PubMed ID: 39494541] partner repository with the dataset identifier PXD066163. Unless otherwise stated, all data supporting the results of this study can be found in the article, supplementary, and source data files. Source data are provided with this paper.

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

## Acknowledgments

S.B. was supported by a fellowship from the Alexander von Humboldt Foundation. The work of S.B and C.S. was supported by the European Research Council under Grant Agreement 101118656 (ERC Synergy project 4D-BioSTEM). The work of S.B. and C.S. was initially supported by the Boehringer Ingelheim Fonds Exploration grant. The authors are grateful to Andreas Brech and Sebastian Schultz from Oslo University Hospital, Oslo, Norway for providing the stable mCherry-p62 RPE1 cell line. The authors would like to thank Sven Gerlach, Bernd Hoffmann and Maithreyan Kuppusamy from Forschungszentrum Jülich, Germany, for help generating the stable miniTurbo-mCherry-p62 RPE1 cell line. The authors would like to thank Terje Johansen from the Arctic University of Norway for the helpful discussions. The authors gratefully acknowledge the electron microscopy access time and computing time granted by the biological EM facility of the Ernst-Ruska Center at Forschungszentrum Jülich. In particular, we thank Julio Ortiz for the recording of the Volta phase plate p62 filament tomograms. The authors gratefully acknowledge the computing time granted by the JARA Vergabegremium and provided on the JARA Partition part of the supercomputer JURECA at Forschungszentrum Jülich[114]. The work of NLK is funded by the Deutsche Forschungsgemeinschaft (DFG, German Research Foundation): EXC 2030–390661388, KO 5091/4-1, DFG-431549029–SFB 1451, and DFG-411422114 - GRK 2550. PFH acknowledges funding by the Deutsche Forschungsgemeinschaft (SFB1381, project-ID 403222702). LT acknowledges the support of the India Alliance DBT/Wellcome Trust grant (IA/21/2/505925).

## Author contributions

S.B., S.M., L.J. and C.S. designed the project. S.M., A.K. and L.J. purified p62 filaments. S.M., A.K. and C.S. determined the cryo-EM structure of p62 with the help of L.S. and L.T. L.J. and A.K. reconstituted p62 filaments with binding partners and LC3b conjugated liposomes. A.K., O.K. and J.F. performed single-particle fluorescence measurements and analysis. SB performed the live cell confocal imaging and data analysis on RPE1, L.I. and N.L.K. prepared mouse samples and performed the fluorescence image analysis. SB performed in situ CLEM followed by cryo-ET analysis. S.B., P.L. and R.D.B. performed the EDX measurements and analysis. S.B., M.M.D. and P.F.H. performed the proximity proteomics experiments and analysis. S.B., L.J., A.K. and C.S. wrote the manuscript with input from the other authors.

## Funding

## Competing interests

The authors declare no competing interests.
