## [Transparent Peer Review file · Nature Communications]

Structural organization of p62 filaments and the cellular ultrastructure of calcium-rich p62-enwrapped lipid droplet cargo

Corresponding Author: Professor Carsten Sachse

Version 0:

Reviewer comments:

Reviewer #1

(Remarks to the Author)

In this manuscript the Sachse laboratory is building on their previous seminal work identifying the formation of p62 filaments by cryo-EM. The improvements in the purification of the full length p62 (which is challenging to say the least) allowed the authors to generate robust filamentous structures and investigate their interactions with LC3B and poly-ubiquitin producing significant impact on the filament length and branching. Most interestingly, the authors also show the formation of a calcium, phosphate, and magnesium (as well as oxygen?)-rich p62 coat enwrapping lipid droplets in cellulo by oligomers, potentially shedding light on the mechanism of action of p62 in the early stages of lipophagy. The study is technologically exciting as the robust filament assembly allowed the authors to visualise how these structures can interact with membrane-conjugated LC3 and ubiquitin, shedding light on their possible behaviour during autophagy initiation. The in situ imaging of p62 and EDX imaging of lipid droplets is intriguing and certainly provides a lot of food for thought, with potentially important (albeit still to be determined) functional consequences.

Major comments:

1. For Figure 2G/H, is there a statistically significant decrease in filament length with LC3b? Although it can be seen that that the modal filament length has shifted from around 200um to 100um, the overall distribution looks fairly similar, so it would be useful to include some statistical analysis to see whether the change is significant.
2. If, as shown in supplementary Figure 1A, the p62 filaments seen at pH 6 are largely unwound at pH 8, what is the physiological relevance of the filament structure described, given that intracellular pH is typically around 7-7.5? Would filament structures at neutral pH interact with LC3/liposomes/ubiquitin in the similar manner to the lower pH?
3. On lines 442-443, the authors state that the control focussing on lipid droplets showing no p62 colocalisation (shown in supplementary Figure 7) obtained a very similar EDX signal. This reads as though the signal is similar to that of the p62-encapsulated lipid droplets, when in fact the droplets without p62 do not show additional Mg, Ca and P signals seen in the p62 coat in Figure 6E. This should be rewritten to clarify that without the p62 coat, the atoms are not detected. More importantly, the mechanism by which p62 drives the formation of the calcium and phosphate coat around LDs is currently not explored. The authors should attempt to address this, either experimentally (can the EDX spectroscopy be applied to the p62 filament preparations in vitro?) or by providing additional discussion exploring possible pathways, including, but not limited to, p62-mediated recruitment of mineralisation factors.
4. Similarly, what is the rationale for attributing the detection of the atoms by EDX spectroscopy to inorganic molecules such as calcium phosphate as opposed to calcium, phosphorus and magnesium being incorporated into proteins within the LD coat? What about calcium-binding and phosphorylated proteins? Is it similar to the oxygen, which is also clearly present within the coat, and which would be assumed to be incorporated into proteins and other biomolecules in the cytoplasm but also on the surface of LDs? Would calcium chelators or phosphatases disrupt the association of respective atoms with LDs? Fully answering these questions might not be possible within this study but the authors should provide a balanced interpretation of their data.
5. What is the connection between the filaments detected in vitro with the p62 coat on LDs? There is no evidence that p62 in situ is polymerised, and beyond the effect of calcium on the filament bundling in vitro there is no support for the filament organisation of p62 on the surface of the LD in the diagram 6I. Will the in situ cryo-EM be able to clarify this issue now or in the future?
6. Likewise, the rationale for including ubiquitinated proteins on LD surfaces in Figure 6I requires clarification. Do the authors have direct evidence that p62 recruitment is driven by ubiquitination? Are LDs displaying p62 ubiquitination-

enriched, while those lacking p62 are not? The authors should discuss known ubiquitination machinery implicated in lipid droplet ubiquitination (doi: 10.1016/j.bbaliip.2017.06.006), and assess whether these might mediate p62 recruitment. This contextualisation would provide a more comprehensive mechanistic perspective.

Minor comments:

7. In line 73 of introduction, the authors refer to the characterisation of other selective autophagy receptors beyond p62 in mitophagy. Various SARs have been identified as cargo receptors across a wide range of autophagy subtypes, so it might be worth referencing a review that provides an overview of this (e.g., Farre & Subramani 2016, doi: 10.1038/nrm.2016.74), rather than just referring to mitophagy alone.

8. When describing domains of p62 in lines 87-89, it is worth noting the disordered linker region between the PB1 and ZZ domains, and its possible role in p62 disulphide-linked conjugate formation (e.g. doi: 10.1038/s41467-017-02746-z).

9. In the second paragraph of the discussion, the authors describe the unwinding and structural opening up of the double-helical filament architecture to form a p62 "bubble" structure, allowing access to binding sites. A diagrammatic figure may be useful here to help the reader visualise what is meant by this.

Reviewer #2

(Remarks to the Author)

The manuscript is very comprehensive but a bit difficult to read because of many abbreviations and different techniques that are presented within the experimental analysis. It also contains a lot of data, probably enough for two papers. Overall, all experiments seem technically sound and according to current standards. The method section is sufficient, only a description for AlphaFold3 usage (especially for creating filaments) is missing.

Reviewer #3

(Remarks to the Author)

Reviewer #4

(Remarks to the Author)

This manuscript "NCOMMS-25-03646" presents a comprehensive structural analysis of full-length p62/SQSTM1, combining state-of-the-art techniques such as high-resolution cryo-electron microscopy single particle analysis (SPA), in situ cryo-electron tomography (cryo-ET), complementary biochemical methods, correlative light and electron microscopy (CLEM), STEM, and EDX. The authors describe a double-helical filament architecture built upon a PB1 domain scaffold with a flexible C-terminal region. They further investigate how binding partners (LC3b and polyubiquitin) and divalent cations influence filament organization. Additionally, the in situ imaging of p62-enwrapped lipid droplets with a calcium phosphate-rich coat offers novel insights into early autophagy stages. Overall, the study is robust and provides valuable structural insights linking molecular architecture to cellular function in autophagy. I recommend acceptance following the consideration of one major and several minor points:

Major:

Low-resolution regions:

Although the PB1 domain scaffold is resolved at ~4.5 Å, the flexible C-terminal and ZZ domains remain poorly defined. These lower-resolution regions are crucial for interpreting binding interactions. Although the authors present AlphaFold predictions and some binding experiment using imaging method, I suggest including additional biochemical characterization to further strengthen these conclusions.

Minor suggestions:

1) It would be beneficial if the authors could explain why the resolution for the full-length structure does not match that previously achieved for the isolated PB1 domain in 2020.

2) An explanation of the differences observed at varying concentrations of p62 filament and GST-4xUbiquitin between Figure 2C (condensates) and Figure 3B (bundles) would be appreciated.

3) Figures 6D and E (right EDX figures) should include scale bars.

4) In Figure S1A, it would be helpful to present the 2D classification results at the same scale across different pH conditions.

5) Typo error: line 1169 (methods, Cryo-electron tomography of reconstituted p62 filaments, line 5) should read "63,000x magnification" instead of "63.000x magnification."

Reviewer #5

(Remarks to the Author)

The authors present cryo-electron microscopy imaging of the selective autophagy receptor p62, revealing that full-length p62 forms regular double-helix filaments in vitro. AlphaFold3 was used to predict the interactions of full-length p62 filaments with LC3b and ubiquitin. The interaction of p62 with soluble LC3b, LC3B-conjugated liposome membranes and poly-ubiquitin was studied. These three interactions led to structural disassembly, lateral association and bundling, respectively, providing insights into how p62 filaments respond to autophagy-related factors.

In vivo experiments showed that p62 puncta colocalized with lipid droplets in cells, with p62 forming a discontinuous coat around the droplets. Elemental analysis of the p62 coat revealed an enrichment of calcium and phosphorus, and the effects of other divalent cations were also examined. The authors argue that these findings support the hypothesis that calcium ions may be a key regulator in autophagy.

While this manuscript presents an improved structural model of p62 filaments, the biological significance of these findings is unclear, and the possible significance for autophagy mechanisms in cells was not clearly stated by the authors. Judging the significance is also made harder by the fact that the context of the present work within the history of this subfield is not clearly spelled out - a general reader would find it difficult to know what precisely has been established in this field, and what has not been established. Further, the in vitro and in vivo experiments are somewhat disconnected.

Major comments:

1. The findings presented in this manuscript are close to previously published findings by the same authors in [Ciuffa et al., 2015] and [Jakobi et al., 2020]. The present study is a worthy extension of these earlier studies; however, the present manuscript's additional impact on the community's understanding of autophagy appears somewhat limited.

2. The authors state that "in contrast to the previously determined PB1 assemblies [Jakobi et al., 2020], the here determined full-length p62 filament structure formed a double helical filament architecture". However, their previous study, [Ciuffa et al., 2015], reported that filaments of PB1-ZZ (a truncated version of full length p62) are organized in a double helix, while full-length p62 filaments consist only of a single strand [Ciuffa et al., 2015]. Thus, the authors own earlier findings appear to contradict their findings in the present work. This is not discussed in the manuscript.

Additionally, Ciuffa et al. described that the C-terminal part of the protein forms a solvent-accessible stretch, a feature that is also highlighted in this manuscript. Furthermore, Ciuffa et al. found that octa-ubiquitin but not LC3 provokes filament shortening, which seems to significantly overlap with the work presented in this manuscript (see section "P62 filament interactions with LC3b and ubiquitin"). The authors should clarify how their findings differ from or extend these previous studies.

3. It would really help if the context of the work presented in this manuscript were clearly described. The manuscript would greatly benefit from a clear exposition, including an explanatory figure, of the current understanding of autophagy as it involves p62, LC3, ubiquitin, Atg8 family proteins, membrane compartments etc etc. For a general reader it is very difficult to comprehend from the present Introduction and other parts of the manuscript.

4. The biological significance of this study is not stated, and no "model" figure is presented that visually announces the authors' hypotheses. What do these findings tell us about the mechanisms of p62-mediated autophagy?

For example, the influence of LC3b and poly-ubiquitin on p62 is studied – but how is this related to autophagy? Another example: in the section "p62 filament interactions with LC3b and ubiquitin", p62 filaments form condensates in the presence of ubiquitin, whereas, in "p62 filament membrane interactions", ubiquitin instead induces parallel filaments bundles. Which of these two behaviors better reflects in vivo conditions? Again, what do these two different outcomes tell us about autophagy? Another example: Ca and p62 are reported to colocalize on lipid droplets in cells – what mechanism involving Ca does this suggest to the authors?

5. The authors emphasize that most p62 filaments have regular helical structure at pH=6, while a fraction unwind at pH=8. However, in the section "p62 filament interactions with LC3b and ubiquitin", the authors polymerized p62 filaments at pH=8. Why is this pH used, when this apparently disrupts filaments?

6. In the section "p62 filament membrane interactions", the authors characterized the ultrastructure of p62 filaments at a high concentration and at pH=6. However, in subsequent experiments, they used a 10-fold lower concentration and pH=7.4. How does this impact the ultrastructure?

7. The authors revealed the structural organization of p62 in human RPE1 cells. However, they did not establish a clear connection between the in vivo and in vitro experiments. Specifically, the physiological concentration of p62 in cells was not reported. Was it comparable to the in vitro conditions? Providing this information would help interpret the relevance of the in vitro findings in the cellular context.

Minor comments:

1. Fig. 2G and 2H show the distribution of p62 filament length in the absence and presence of LC3b, suggesting that LC3b induces depolymerization. For a clearer comparison, plotting both distributions within the same graph would be more effective.

2. In Fig. S7F, the label for the peak at around $e = 500$ eV is missing.

3. The authors compare the p62 structure to the DNA double-helix model (Line 494). However, given the subunit composition and the pitch of the p62 double helix filaments, perhaps a comparison to actin filaments or microtubules might be more appropriate.

Version 1:

Reviewer comments:

Reviewer #1

(Remarks to the Author)

The authors performed extensive revisions which significantly improved the manuscript. The new proteomics data adds to the proposed working model of the p62/Ca²⁺ coat formation. Other concerns from this reviewers were also constructively addressed. As such, I would recommend the manuscript for publication in the current form.

I would still recommend spell check of the text and figures, e.g. Holliday junctions in one of the figures is spelled with one L. "In cellulo" should be spelled "in cellula" according to Latin grammar.

I've read the Reviewer 5 comments and responses by the authors and I think they are satisfactory. It is impossible for me to know if the Reviewer would accept them, but from the extensive experimental and textual revisions I would say most or all of the criticisms have been addressed. The authors did a very nice job in providing clarifications requested by the reviewer and formalising their model further. Undoubtedly, there are many uncertainties about the role of p62 and inorganic molecules at LDs but the data is robust, technology is cutting edge, and scientific advance and significance for the autophagy field and cell biology in general is obvious to me.

Reviewer #2

(Remarks to the Author)

The revised manuscript is fundamentally o.k.. It is in my opinion too long, however. But this is an editorial decision.

Reviewer #3

(Remarks to the Author)

Reviewer #4

(Remarks to the Author)

The revision substantially strengthens the manuscript and addresses most of my previous comments. I recommend acceptance pending minor revisions and one concern.

Concern:

1) The authors report a total dose of $500 \text{ e}^-/\text{Å}^2$ for the EDX measurements. I am concerned that the dose might be too high, which caused too much damage to the sample.

Minor revisions

2) Total tomography dose: Please recalculate and confirm the cumulative dose; the per-tilt values reported in Methods (lines 1030–1031 and 1233–1235) imply a total dose of $\sim 80 \text{ e}^-/\text{Å}^2$.

3) Typo: Methods, "Cryo-electron tomography of reconstituted p62 filaments," line 5 (around line 1030) should read "63,000× magnification" (not "63.000×").

4) Supplementary figure quality: Supplementary Figures 7–9 are more blurred in the current merged file than in the previous version. Please check the file format/resolution and restore the prior quality.

Preliminary Response to Reviewer Comments

We thank the reviewers for their detailed comments on the manuscript. In this point-by-point response, we addressed the remaining raised concerns.

In summary, we added two major experiments in this revised version of the manuscript:

1. Proteomic characterization of ATG5 phenotype by p62 proximity labeling
2. Extend biological significance of ATG5 knockdown phenotype to primary cortical neurons and brain cells by fluorescence characterization

The included experiments extend the scope of the conclusions. The remaining questions, e.g. regarding the experimental pHs, were succinctly addressed by amending the presentation in the figures as well as adding introductory and discussion items. Noteworthy, we added a clear graphical model in Figure 7E relevant for the context of the autophagy process.

Reviewer #1 (Remarks to the Author):

In this manuscript the Sachse laboratory is building on their previous seminal work identifying the formation of p62 filaments by cryo-EM. The improvements in the purification of the full length p62 (which is challenging to say the least) allowed the authors to generate robust filamentous structures and investigate their interactions with LC3B and poly-ubiquitin producing significant impact on the filament length and branching. Most interestingly, the authors also show the formation of a calcium, phosphate, and magnesium (as well as oxygen?)-rich p62 coat enwrapping lipid droplets in cellulo by oligomers, potentially shedding light on the mechanism of action of p62 in the early stages of lipophagy. The study is technologically exciting as the robust filament assembly allowed the authors to visualise how these structures can interact with membrane-conjugated LC3 and ubiquitin, shedding light on their possible behaviour during autophagy initiation. The in situ imaging of p62 and EDX imaging of lipid droplets is intriguing and certainly provides a lot of food for thought, with potentially important (albeit still to be determined) functional consequences.

Thank you for the overall positive assessment of our manuscript.

Major comments:

1. For Figure 2G/H, is there a statistically significant decrease in filament length with LC3b? Although it can be seen that that the modal filament length has shifted from around 200um to 100um, the overall distribution looks fairly similar, so it would be useful to include some statistical analysis to see whether the change is significant.

Stimulated by the reviewer's comments, we added an extended statistical treatment on the filament length. Interestingly, the filament length distribution does not follow a Gaussian distribution. Therefore, we performed the analysis using an empirical cumulative density function: Details have been added to the legend of new Supplementary Fig. 11B and the Methods section of the manuscript (page 40 (Supplementary Fig. 11B), 58):

The cumulative density distribution (CDF) was calculated for p62 filaments without LC3b (blue line) and p62 filaments in the presence of LC3b (orange line). The dashed red line gives the maximum absolute difference between the two CDFs, $D=0.17787$. The Kolmogorov-Smirnov test statistic was calculated $K= 2.10023$ and the approximate p-value was derived $p= 0.00014746$. Since $p<0.05$ the observed shift of the distribution in the presence of LC3b towards shorter filament lengths is statistically significant.

2a. If, as shown in supplementary Figure 1A, the p62 filaments seen at pH 6 are largely unwound at pH 8, what is the physiological relevance of the filament structure described, given that intracellular pH is typically around 7-7.5?

We thank the reviewer for asking this question. We address this point in a dedicated section

in the discussion (page 26):

In the cryo-EM images at physiological pH 7-8, we also found an unwinding and structural opening up of the double-helical filament architecture located on the membrane surface. In contrast, our full length p62 filament structure was solved at pH 6 in order to obtain a more homogeneous closed filament conformation following the helical symmetry. At a cytosolic pH of approximately 7.4, the p62 filaments reveal the ability to adopt an open “assembly configuration”, in which previously inaccessible available binding sites in the interior will become accessible in an opened p62 “bubble” structure while still maintaining functional polymers and oligomers. At the slightly lower pH values present in the fully assembled autophagosome, estimated to be closer to pH 6, p62 will assemble in more rigid filaments. In future work, it will be interesting to characterize LC3 and poly-ubiquitin binding at various pH values to better understand this regulatory mechanism, similar to the mechanism described for arginylated substrate⁴⁵.

b. Would filament structures at neutral pH interact with LC3/liposomes/ubiquitin in the similar manner to the lower pH?

During the restricted revision period, we attempted to perform quantitative biochemical binding studies at different pHs. Unfortunately, the experiments have not been conclusive. Please find the comment on biochemical bulk characterization of p62 filaments requested by reviewer #4, point 2.

3. On lines 442-443, the authors state that the control focussing on lipid droplets showing no p62 colocalisation (shown in supplementary Figure 7) obtained a very similar EDX signal. This reads as though the signal is similar to that of the p62-encapsulated lipid droplets, when in fact the droplets without p62 do not show additional Mg, Ca and P signals seen in the p62 coat in Figure 6E. This should be rewritten to clarify that without the p62 coat, the atoms are not detected.

We apologize for the misunderstanding and added a clarifying statement:

As a control, we focused on lipid droplets showing no p62 co-localization and obtained a very similar EDX signal of the core lipid droplet region while lacking the high levels of Ca and P in their surrounding (Supplementary Figure 8).

4. More importantly, the mechanism by which p62 drives the formation of the calcium and phosphate coat around LDs is currently not explored. The authors should attempt to address this, either experimentally (can the EDX spectroscopy be applied to the p62 filament preparations in vitro?) or by providing additional discussion exploring possible pathways, including, but not limited to, p62-mediated recruitment of mineralisation factors.

The reviewer asks an intriguing question. To address the raised point, we turned to Turbo-ID proximity labeling followed by subsequent proteomic mass spectrometry to identify the molecular protein composition and differences in p62 proximity with regards to wildtype and ATG5 K.D. lipid droplets (page 22, new Figure 7A-D):

In order to provide further molecular insights in the observed p62-encapsulated lipid droplets, we turned to a proximity proteomics experiment⁶¹. In short, the miniTurbo enzyme was fused to mCherry-p62 and a stable RPE1 cell line was generated. Subsequently, proteins in the immediate proximity of p62 were biotinylated, pulled down and identified using mass spectrometry (**Figure 7A/B**). The abundance of biotinylated proteins was estimated for the 72-hour ATG5 knockdown (including mCherry-p62 and miniTurbo), i.e. T-mCh-p62 ATG5 KD, and compared with a control of endogenously expressed ATG5 (including mCherry-p62 and miniTurbo), i.e. T-mCh-p62, as well as a second control devoid of the miniTurbo enzyme (including mCherry-p62), i.e. mCh-p62. Western blot confirmed the absence of ATG5 in the T-mCh-p62 ATG5 KD sample while

ATG5 was detected in the mCh-p62 and T-mCh-p62 control samples (**Figure 7C**). Consistently, we found neither ATG5 nor ATG16 in the proteomic data of the T-mCh-p62 ATG5 KD sample (ProteomeXchange-ID PXD066163). We detected several known interactors of p62 and other early autophagy proteins in both the T-mCh-p62 and T-mCh-p62 ATG5 KD samples: NBR1, ATG13, ZFYVE1/DFCP1 and TRAF6, while they were not present in the mCh-p62 control. Statistical analysis of relative peptide abundance between the T-mCh-p62 ATG5 KD and the T-mCh-p62 endogenous ATG5 sample identified several other proteins related to autophagy that were only present in the absence of ATG5 (**Figure 7D**). Most interestingly is the presence of SIRT1 and alpha-synuclein as these proteins have been described to be present on lipid droplets involved in lipophagy^{62,63}. Analysis of protein enrichment and depletion between endogenous ATG5 and ATG5 knockdown also identified proteins related to lipid droplets or lipid and steroid metabolism: TPRG1L⁶⁴, SQLE and PIP4P1. Finally, we inspected proteins that were possibly related to the observed calcium accumulation around the p62-lipid droplet structures. Intracellular calcium chelators, such as calmodulin or calcineurin, were neither enriched nor depleted upon ATG5 knockdown. However, we identified a calcium channel and a regulator of calcium homeostasis in the ATG5 knockdown but not in the endogenous ATG5 sample: First, ITPR2, an ER-resident, inositol 1,4,5-trisphosphate-gated calcium channel and, second, MCUb known as a negative regulator for the mitochondrial calcium uniporter. The former proteins, i.e. ITPR channels, release calcium from the ER and have been shown affect autophagy⁶⁵. The latter protein, i.e. MCUb, has been described to block calcium overload of nearby mitochondria, and also been associated with the early stages of autophagy and mitophagy⁶⁶⁻⁶⁸. Together, our proximity proteomics experiments on the ATG5 knockdown induced p62-encapsulated lipid droplets revealed proteins such as SIRT1 and alpha-synuclein, linking lipid droplets to autophagy. Moreover, upon ATG5 knockdown proteins involved in calcium transport from the ER such as ITPR2 and MCUb were found in close proximity to p62 supporting the notion that the observed calcium layer and nearby single phospholipid bilayer may represent calcium released from the ER.

We also highlight the role of identified ER calcium transporters in the discussion on page 30: The fact that we identified calcium transporters in our proteomics studies, such as ITPR2, further lends support to this hypothesis. ITPR2, an inositol 1,4,5-trisphosphate-gated calcium channel found in the ER membrane, has been shown to be involved in autophagy initiation and senescence⁶⁶. Additional reports demonstrated that knockdown of its paralogue, ITPR1, led to autophagosome accumulation⁹², and ITPR2/ITPR3 knockdown reduced FIP200 punctae, WIPI2 punctae and LC3 punctae under starvation conditions⁹⁰.

5. Similarly, what is the rationale for attributing the detection of the atoms by EDX spectroscopy to inorganic molecules such as calcium phosphate as opposed to calcium, phosphorus and magnesium being incorporated into proteins within the LD coat? What about calcium-binding and phosphorylated proteins? Is it similar to the oxygen, which is also clearly present within the coat, and which would be assumed to be incorporated into proteins and other biomolecules in the cytoplasm but also on the surface of LDs? Would calcium chelators or phosphatases disrupt the association of respective atoms with LDs? Fully answering these questions might not be possible within this study but the authors should provide a balanced interpretation of their data.

For the response, please refer to point 4 above. The performed proximity labeling addresses the question.

6. What is the connection between the filaments detected *in vitro* with the p62 coat on LDs? There is no evidence that p62 *in situ* is polymerised, and beyond the effect of calcium on the filament bundling *in vitro* there is no support for the filament organisation of p62 on the surface of the LD in the diagram 6I.

We thank the reviewer for pointing out this question and added our state of understanding to the discussion (page 29):

An intriguing question arises whether the here observed double-helical filaments from purified material are compatible with our cryo-ET observations in cells. Given the dense cellular environment and limited resolution of the lamellae we could not find direct visual evidence of the helically assembled filaments. Nevertheless, we located p62 surrounding the lipid droplet as part of a calcium-rich layer and believe it to be present as structurally modified polymeric assemblies like the ones visualized on the liposome surface or our *in vitro* reconstituted calcium-induced filament bundles (see **Figure 3D and 6H**).

7. Will the *in situ* cryo-EM be able to clarify this issue now or in the future?

As actin filaments (7 nm width) are routinely visualized in vitrified cells by cryo-ET, the currently achievable resolution is sufficient to locate such structures in cells. Therefore, at this moment it appears that the chosen ATG5 knockdown conditions do not favor the double-helical filament assembly of p62 while they are compatible with polymeric assemblies as part of the calcium-rich layer (refer to point 6 above).

8. Likewise, the rationale for including ubiquitinated proteins on LD surfaces in Figure 6I requires clarification. Do the authors have direct evidence that p62 recruitment is driven by ubiquitination? Are LDs displaying p62 ubiquitination-enriched, while those lacking p62 are not? The authors should discuss known ubiquitination machinery implicated in lipid droplet ubiquitination (doi: 10.1016/j.bbaliip.2017.06.006), and assess whether these might mediate p62 recruitment. This contextualisation would provide a more comprehensive mechanistic perspective.

We attempted to address this contextualization question and discuss our proteomic results in the manuscript (page 27):

However, without an additional ubiquitin enrichment step we were unable to experimentally confirm that PLIN ubiquitination recruited p62 to these lipid droplets, and we did not identify other proteins of the lipid droplet ubiquitination machinery⁷⁶ in our proteomic mass spectrometry experiments.

Minor comments:

9. In line 73 of introduction, the authors refer to the characterisation of other selective autophagy receptors beyond p62 in mitophagy. Various SARs have been identified as cargo receptors across a wide range of autophagy subtypes, so it might be worth referencing a review that provides an overview of this (e.g., Farre & Subramani 2016, doi: 10.1038/nrm.2016.74), rather than just referring to mitophagy alone.

We added the review reference as requested.

8. When describing domains of p62 in lines 87-89, it is worth noting the disordered linker region between the PB1 and ZZ domains, and its possible role in p62 disulphide-linked conjugate formation (e.g. doi: 10.1038/s41467-017-02746-z).

We added the reference and mentioned the disulphide-linked conjugate as requested.

9. In the second paragraph of the discussion, the authors describe the unwinding and structural opening up of the double-helical filament architecture to form a p62 “bubble” structure, allowing access to binding sites. A diagrammatic figure may be useful here to help

the reader visualise what is meant by this.

Stimulated by the reviewer's comment, in order to help the discussion on this item, we included a little cartoon of this bubble (Supplementary Fig. 11A).

Reviewer #2 (Remarks to the Author):

1. The manuscript is very comprehensive but a bit difficult to read because of many abbreviations and different techniques that are presented within the experimental analysis. It also contains a lot of data, probably enough for two papers. Overall, all experiments seem technically sound and according to current standards.

Thank you for the overall positive assessment of the manuscript. We firmly believe that different techniques are essential to comprehensively characterize the structure and function of this important autophagy receptor p62/SQSTM1. We made sure that the used abbreviations are not excessive while respecting the standards of the respective fields. We will keep them restricted to ascertain readability while they are transparently introduced at the beginning of the manuscript.

2. The method section is sufficient, only a description for AlphaFold3 usage (especially for creating filaments) is missing.

According to the request, we added a dedicated methods section (page 43) in addition to the previously presented supplementary Figure S2:

To model the p62 filament structure, we used AlphaFold3 (AF3) multimer predictions (Abramson et al., 2024): first we tested p62 full-length sequences arranged in dimers, trimers, tetramers up to decamers. AF3 consistently favored closed ring-like assemblies, even for small oligomers (≥ 3 subunits) while for larger rings the resulting p62 rings exhibited close-to-rotational symmetry. A single p62 subunit was excised from a AF3-generated p62 closed ring-like decamer with a diameter of 145 Å approaching the diameter of the filament. To relate these predictions to the p62 filament, we fitted the PB1 domain in the corresponding well-resolved density of one strand of the filament map while including the full-length p62 model and repeated the operation for the opposite PB1 strand. Using the determined helical symmetry parameters of 11.0 Å helical rise and a -26.1° helical rotation, we propagated the two subunits along the helical axis, generating a double-helical filament model composed of repeating p62 units. This helical assembly yielded a plausible model of the p62 filament, with subunits well fitting the experimental cryo-EM density while maintaining interaction geometries close to the AF3-predicted oligomers.

Reviewer #4 (Remarks to the Author):

This manuscript "NCOMMS-25-03646" presents a comprehensive structural analysis of full-length p62/SQSTM1, combining state-of-the-art techniques such as high-resolution cryo-electron microscopy single particle analysis (SPA), in situ cryo-electron tomography (cryo-ET), complementary biochemical methods, correlative light and electron microscopy (CLEM), STEM, and EDX. The authors describe a double-helical filament architecture built upon a PB1 domain scaffold with a flexible C-terminal region. They further investigate how binding partners (LC3b and polyubiquitin) and divalent cations influence filament organization. Additionally, the in situ imaging of p62-enwrapped lipid droplets with a calcium phosphate-rich coat offers novel insights into early autophagy stages. Overall, the study is robust and provides valuable structural insights linking molecular architecture to cellular function in autophagy. I recommend acceptance following the consideration of one major and several minor points:

We thank the reviewer for the overall positive assessment of the manuscript.

Major:

1. Low-resolution regions:

Although the PB1 domain scaffold is resolved at ~4.5 Å, the flexible C-terminal and ZZ domains remain poorly defined. These lower-resolution regions are crucial for interpreting binding interactions.

For clarification: the limited resolution of the densities C-terminal to the ZZ-domain is due to molecular flexibility (complete or partial structural disorder). Such structures cannot be resolved by typical averaging based structural methods such as cryo-EM in particular when they do not follow the imposed helical symmetry. We added a clarifying sentence in the results section (page 6):

The densities of the associated C-terminal domains were weak and poorly resolved as they did not follow helical symmetry presumably due to molecular flexibility with respect to the PB1 strands, which is in agreement with the assigned structurally disordered region in residues 169-338²¹.

We added another sentence to the discussion (page 25):

The density located C-terminally to the ZZ-domain could only be resolved at lower resolution as it does not strictly follow the helical symmetry of the PB1 scaffold. An additional reason for the limited resolution is likely the at least partial structural disorder of the region 168 – 338 in agreement with previous disorder predictions²¹ also reflected by the lower confidence score of the AF3 predictions included in our modeled p62 filament.

2. Although the authors present AlphaFold predictions and some binding experiment using imaging method, I suggest including additional biochemical characterization to further strengthen these conclusions.

We thank the reviewer for the suggestion. In the light of the reviewer's comment, we wanted to share that we performed proposed biochemical characterization for full-length p62 with LC3 and ubiquitin using surface plasmon resonance (SPR) and bilayer interferometry (BLI), however, without success. We believe that the results were not conclusive due to the aggregation-prone nature of full-length p62. Once full-length p62 is freed from its MPB-tag, p62 maintains an immediate assembly/disassembly equilibrium. In this context, we emphasize this point in the discussion section of the manuscript (page 25):

Previous studies have successfully employed bulk structural and biochemical characterization methods to characterize p62 interactions but only for relevant p62 fragments, e.g. with the isolated PB1 domain, LIR motif peptide and UBA domain^{12,70,71} but not for full-length p62 investigated here. When working with full-length p62, bulk characterization methods are often not conclusive due to the contributions of p62 self-interactions and associated heterogeneity. Therefore, we turned to single-molecule structure characterization methods such as negative staining and cryo-EM as well as single-particle fluorescence microscopy that allow targeted identification of the relevant subpopulations for the interactions.

Minor suggestions:

3. It would be beneficial if the authors could explain why the resolution for the full-length structure does not match that previously achieved for the isolated PB1 domain in 2020.

Regarding the resolution, please refer to the response of point 1.

We highlight the structural differences between the isolated PB1 and full-length p62 assemblies more clearly in the discussion (page 25):

In contrast, the more recently determined PB1 (1-102) assemblies²² acquired on direct electron detectors allowed reliable atomic model building revealing triple or quadruple

PB1 domain strands leading to a closed tube. In comparison, the here determined full-length structure revealed only two well-resolved antiparallel PB1 scaffold strands resulting in an open major groove while the previously determined PB1 domain-only assemblies had one or two additional PB1 strands inserted into the closed helical assembly. In the full-length structure, the space required by the additional 340 C-terminal amino acids restricts the assembly to two antiparallel PB1 strands leaving space for a major groove.

4. An explanation of the differences observed at varying concentrations of p62 filament and GST-4xUbiquitin between Figure 2C (condensates) and Figure 3B (bundles) would be appreciated.

The differences arose primarily from different experimental requirements in the single-molecule TIRF microscopy and the cryo-EM studies. We added a clarifying discussion on page 25:

As a consequence of the chosen single-molecule characterization methods, for successful detection of the relevant interactions we also had to adjust the p62 working concentrations to nM and μ M for single-particle fluorescence microscopy/negative staining EM and cryo-EM, respectively. These concentration differences also explain the differences in the observed fine p62 structures after incubation with GST-4xUb, e.g. between negative staining EM and cryo-EM (see **Figure 2C and 3B**). The formation of long filaments is favored at μ M over the much shorter p62 fragments at nM concentration while both morphologies are consistent with the formation of cross-linked larger p62 assemblies.

3) Figures 6D and E (right EDX figures) should include scale bars.

We added scale bars as requested.

4) In Figure S1A, it would be helpful to present the 2D classification results at the same scale across different pH conditions.

In order to address this point, we updated the class averages at the same scale.

5) Typo error: line 1169 (methods, Cryo-electron tomography of reconstituted p62 filaments, line 5) should read “63,000x magnification” instead of “63.000x magnification.”

Thank you for spotting the typo.

Reviewer #5 (Remarks to the Author):

The authors present cryo-electron microscopy imaging of the selective autophagy receptor p62, revealing that full-length p62 forms regular double-helix filaments in vitro. AlphaFold3 was used to predict the interactions of full-length p62 filaments with LC3b and ubiquitin. The interaction of p62 with soluble LC3b, LC3B-conjugated liposome membranes and poly-ubiquitin was studied. These three interactions led to structural disassembly, lateral association and bundling, respectively, providing insights into how p62 filaments respond to autophagy-related factors.

In vivo experiments showed that p62 puncta colocalized with lipid droplets in cells, with p62 forming a discontinuous coat around the droplets. Elemental analysis of the p62 coat revealed an enrichment of calcium and phosphorus, and the effects of other divalent cations were also examined. The authors argue that these findings support the hypothesis that calcium ions may be a key regulator in autophagy.

While this manuscript presents an improved structural model of p62 filaments, the biological significance of these findings is unclear, and the possible significance for autophagy mechanisms in cells was not clearly stated by the authors. Judging the significance is also made harder by the fact that the context of the present work within the history of this subfield is not clearly spelled out - a general reader would find it difficult to know what precisely has been established in this field, and what has not been established. Further, the *in vitro* and *in vivo* experiments are somewhat disconnected.

We thank the reviewer for taking the time to review the manuscript in detail.

Major comments:

1. The findings presented in this manuscript are close to previously published findings by the same authors in [Ciuffa et al., 2015] and [Jakobi et al., 2020]. The present study is a worthy extension of these earlier studies; however, the present manuscript's additional impact on the community's understanding of autophagy appears somewhat limited.

We respectfully disagree with the conclusion of this reviewer. Ciuffa et al. established that helical filaments of p62 are formed *in vitro* by low-resolution cryo-EM (see more details on point 2 below) while Jakobi et al. focused the structural studies on the close-to-atomic resolution structure of the p62-PB1 domain and its requirements in cells. The current study reveals state-of-the-art structural results and addresses, first, the full-length p62 three-dimensional double-helical organization (that was not studied previously) and second, for the first time, by high-resolution cryo-EM *in cellulo* of how p62 encapsulates lipid droplets during lipophagy and discovers an unexpected role in divalent cation p62 binding. In point 3, this reviewer requests a model for better understanding of the autophagy process. We are happy to provide and highlight these novel details in the revised version of the manuscript.

2. The authors state that “in contrast to the previously determined PB1 assemblies [Jakobi et al., 2020], the here determined full-length p62 filament structure formed a double helical filament architecture”. However, their previous study, [Ciuffa et al., 2015], reported that filaments of PB1-ZZ (a truncated version of full length p62) are organized in a double helix, while full-length p62 filaments consist only of a single strand [Ciuffa et al., 2015]. Thus, the authors own earlier findings appear to contradict their findings in the present work. This is not discussed in the manuscript.

We believe that this point (as well as related point 1) most likely arose from a lack of technical understanding of the associated structural EM data quality in our previous Ciuffa et al., Jakobi et al. and the current manuscripts. Therefore, we explain the different structural findings in the discussion in comparison (page 25):

Our own earlier work of full-length p62 revealed the ability to form filamentous assemblies while the cryo-EM images were acquired on a CCD camera and single as well as double-helical filaments could not be discerned unambiguously at the limited resolution²¹. In contrast, the more recently determined PB1 (1-102) assemblies²² acquired on direct electron detectors allowed reliable atomic model building revealing triple or quadruple PB1 domain strands leading to a closed tube. In comparison, the here determined full-length structure revealed only two well-resolved antiparallel PB1 scaffold strands resulting in an open major groove while the previously determined PB1 domain-only assemblies had one or two additional PB1 strands inserted into the closed helical assembly.

In this manuscript, we unambiguously determined the double-helical architecture of full-length p62 at 4.5 Å resolution and deposited the PDB coordinates in the protein data bank fully transparently representing state-of-the-art cryo-EM standards.

3. Additionally, Ciuffa et al. described that the C-terminal part of the protein forms a solvent-accessible stretch, a feature that is also highlighted in this manuscript. Furthermore, Ciuffa et al. found that octa-ubiquitin but not LC3 provokes filament shortening, which seems to significantly overlap with the work presented in this manuscript (see section “P62 filament interactions with LC3b and ubiquitin”). The authors should clarify how their findings differ from or extend these previous studies.

We understand the confusion that may have arisen from comparing the two studies. We wish to clarify this confusion briefly for the reviewer:

In Ciuffa et al., we included two basic binding experiments simply mixing p62 filaments with LC3 and octa-ubiquitin separately and reported co-sedimentation and negatively stained micrographs. Here, we significantly go beyond the earlier work, perform single-particle fluorescence microscopy to record the association over time and show a comprehensive analysis of the interactors including their mixtures (ubiquitin + LC3) mimicking condensate-forming conditions and perform experiments in the presence of membrane-conjugated LC3. Moreover, we explore the importance of different concentrations highly relevant for these interactions (refer to point reviewer #4, point 4). Furthermore, we spent significant efforts to analyze the associated fine structures of p62 assemblies by cryo-electron tomography at unprecedented detail including image analysis of segmentation to characterize the molecular morphologies of the observed condensates and filament bundles. Given this comprehensive set of new experimental setups, we cannot fully understand the conclusion of this reviewer.

3. It would really help if the context of the work presented in this manuscript were clearly described. The manuscript would greatly benefit from a clear exposition, including an explanatory figure, of the current understanding of autophagy as it involves p62, LC3, ubiquitin, Atg8 family proteins, membrane compartments etc etc. For a general reader it is very difficult to comprehend from the present Introduction and other parts of the manuscript.

We understand the reviewer’s concern. While we cannot dedicate an entire Figure 1 just summarizing previous results, we extended the introduction text with regards to that comment (page 3): “At the early stages of autophagy, selective autophagy receptors like p62 thereby bridge ubiquitinated cargo via conjugated Atg8 to the forming autophagosome.” We also added a minimal panel to Figure 1A including the primary structure and binding partners.

4. The biological significance of this study is not stated, and no “model” figure is presented that visually announces the authors’ hypotheses. What do these findings tell us about the mechanisms of p62-mediated autophagy?

We thank the reviewer for pointing this out. We addressed the wider biological mechanism by two far-reaching experiments: First, we extended the analysis of ATG5 knockout with respect to p62 to primary neurons and brain cells. Second, we initiated a proteomic analysis of the p62 proximity labeled cells (refer to response of reviewer #1, point 4).

First, we initiated a study in an alternative primary neuron cell line as well as cortical neurons with ATG5 knocked out and confirmed the wider presence of the described p62 positive lipid droplet structures in cells using fluorescence microscopy (page 14):

In order to verify that our observations were not limited to adherent RPE1 cells, we extended our experiments to mouse primary neurons and brain cortical tissue. First, we isolated primary neuronal cells from conditional *Atg5^{flox}:CAG-Cre^{Tmx}* newborn mice, a model previously shown to exhibit robust and complete knockout (KO) of ATG5⁵⁵. After fixation, we labeled WT and KO astrocytes and neurons with an antibody against p62, and stained lipid droplets with Lipidspot488 (**Figure 4D-G**). Second, we analyzed cortical brain sections from three-month-old WT and conditional ATG5 (cKO) mice, in which ATG5 deletion is driven by the neuronal-specific *Sc132a1* promoter⁵⁶. These sections were stained with antibodies against p62 and Perilipin1, a protein associated

with the surface of lipid droplets (**Figure 4H-I**). We used confocal microscopy to image these samples and quantified the amount of lipid droplets per cell, the amount of p62 punctae per cell, the average size of the lipid droplets and the percentage of lipid droplets that co-localized with p62 signal in each cell. Both in human and mouse cells, we observed an increase in the amount of lipid droplets, while this effect was most pronounced in mouse primary KO astrocytes and cortical 3-month-old KO neurons (**Figure 4J**). Additionally, as expected, we found a significant increase in the number of p62 punctae upon ATG5 KD or KO in the analyzed cells (**Figure 4K**). In the absence of ATG5, the average size of the lipid droplets increased from 0.3 to 0.4 μm^2 , from 0.3 to 0.6 μm^2 , from 0.7 to 0.8 μm^2 , for mCherry-p62 RPE1, mouse astrocytes and neurons, respectively (**Figure 4L**). The mouse cortical neurons cells were harder to quantify as they, on average, contained few, very large lipid droplets. Finally, we also detected a significant increase in the number of lipid droplets that co-localized with p62 punctae in those samples lacking ATG5 (**Figure 4M**).

In the discussion on page 27, we highlight:

We observed more and larger lipid droplets in our ATG5 knockdown cells, as well as a higher percentage of lipid droplets co-localized with p62. We confirmed that this is not a unique phenotype limited to human RPE1 cells, as we observed the same trends in mouse ATG5 KO primary astrocytes, mouse primary neurons and mouse cortex (see **Figure 4**).

5a. For example, the influence of LC3b and poly-ubiquitin on p62 is studied – but how is this related to autophagy?

In the paragraphs on page 3 and 4 of the manuscript, we openly introduced how LC3b and poly-ubiquitin binding are relevant to autophagy progression and point to the critical references.

b. Another example: in the section “p62 filament interactions with LC3b and ubiquitin”, p62 filaments form condensates in the presence of ubiquitin, whereas, in “p62 filament membrane interactions”, ubiquitin instead induces parallel filaments bundles. Which of these two behaviors better reflects in vivo conditions? Again, what do these two different outcomes tell us about autophagy?

The observed differences are related to the chosen different experimental conditions. It was our aim to recapitulate the key binding events of selective autophagy in the test tube – a common strategy of biochemical studies also known as bottom-up. First, we mixed p62 filaments and LC3 and GST-4xUb in solution concluding correctly: “filaments form condensates in the presence of ubiquitin”. Second, we mixed p62 filaments with LC3-conjugated liposomes and GST-4xUb and conclude the formation of filament bundles. In the second experiment, we are obviously closer to the cellular situation as LC3 is conjugated to lipid bilayer membrane. All of the studies aim to study the fine structure of p62 assemblies upon addition of the relevant molecular components involved in the early events of selective autophagy. While both outcomes are somewhat different, both have common features: the recruitment of key binding partners of LC3 and Ubiquitin to polymeric p62 assemblies.

c. Another example: Ca and p62 are reported to colocalize on lipid droplets in cells – what mechanism involving Ca does this suggest to the authors?

We laid out our findings and showed for the first time the detailed characterization of the ATG5 knockdown condition. Surprisingly, we discover a role for calcium together with p62 in a layer surrounding lipid droplet cargo. We realize that we have not concluded so clearly on a role in autophagy. We now present a model of how the calcium-rich p62 layer surrounding a lipid droplet may be involved in the early stages of selective autophagy (Figure 7E, page 30):

Given the multitude of observations in a series of biochemical and cellular experiments, we summarize our findings in a model of the early stages of selective autophagy (**Figure**

7E): p62 polymers recognize the ubiquitinated surface of lipid droplets destined for degradation in proximity to the ER. In parallel, an unknown signal leads to calcium efflux from the ER that is collected by the p62 polymers surrounding the cargo with a calcium-rich layer. Additional autophagy factors are recruited leading to the progression of autophagy, e.g. the ATG5/12/16 complex conjugating LC3b to the phosphatidyl ethanolamine (PE) membrane and thereby mediating phagophore formation and extension up to an autophagosome. In the absence of ATG5 membrane extension, LC3b is not conjugated to PE leading to an accumulation of the calcium-rich p62 positive lipid droplet cargo structures.

6. The authors emphasize that most p62 filaments have regular helical structure at pH=6, while a fraction unwind at pH=8. However, in the section “p62 filament interactions with LC3b and ubiquitin”, the authors polymerized p62 filaments at pH=8. Why is this pH used, when this apparently disrupts filaments?

This point was also independently raised by Reviewer #1, point 2. Find the response above. We clarified our rationale in the respective results sections:

- a) On page 5 rationale for cryo-EM structure determination including new Supplementary Figure 1A: “As previous studies suggested an effect of pH on the assembly conditions⁴⁵, we compared the preparations at different pHs ranging from 6 to 8 (Supplementary Figure 1A).”
- b) On page 8 rationale for binding studies: “We used the above-characterized preparations of p62 filaments at pH=8 as they were closer to physiological pH than the stable pH 6 filaments formed for structural analysis.”

7. In the section “p62 filament membrane interactions”, the authors characterized the ultrastructure of p62 filaments at a high concentration and at pH=6. However, in subsequent experiments, they used a 10-fold lower concentration and pH=7.4. How does this impact the ultrastructure?

This point on different experimental concentrations was independently raised by Reviewer #4, point 4. Primarily, they are restricted by the employed experimental characterization method. Therefore, we took care to report them transparently. Refer to the response above.

8. The authors revealed the structural organization of p62 in human RPE1 cells. However, they did not establish a clear connection between the *in vivo* and *in vitro* experiments. Specifically, the physiological concentration of p62 in cells was not reported. Was it comparable to the *in vitro* conditions? Providing this information would help interpret the relevance of the *in vitro* findings in the cellular context.

Relating the *in vivo* concentrations with the *in vitro* experiments is an intriguing question that we included to address in the discussion on page 28:

As we learnt from our *in vitro* experiments, the tendency of p62 filament assembly/disassembly and structural properties such as filament length is dependent on the concentrations present in the test tube. In this context, it is an important question what the cellular concentration of p62 is. Previous reports estimated for mouse fibroblast cells and HEK293T cells the average p62 copy number to be 25,000 and 97,000, respectively^{82,83}. Assuming an average cytoplasmic volume of 1500 μm^3 of an RPE1 cell⁸⁴, the resulting average concentration is between 0.1 μM /100 nM and 0.4/400 nM. This concentration is about three orders of magnitude lower than the 2 mg/ml (41.9 μM) used in this study to generate filaments for structural analysis. While the 100 nM concentrations are of the same order of magnitude as used in the nM binding studies examined by single-particle fluorescence microscopy and negative staining EM. Nevertheless, in the cell we expect significant variation in local concentrations, e.g. when p62 forms a condensate or a dense layer around a lipid droplet cargo, making

significant local structural enrichment and effective higher concentrations possible.

Minor comments:

1. Fig. 2G and 2H show the distribution of p62 filament length in the absence and presence of LC3b, suggesting that LC3b induces depolymerization. For a clearer comparison, plotting both distributions within the same graph would be more effective.

We agree with the reviewer and made the two graphs more comparable including the statistical significance tests that were requested by point1 of reviewer #1.

2. In Fig. S7F, the label for the peak at around $e = 500$ eV is missing.

Added as requested.

3. The authors compare the p62 structure to the DNA double-helix model (Line 494). However, given the subunit composition and the pitch of the p62 double helix filaments, perhaps a comparison to actin filaments or microtubules might be more appropriate.

The comparison to the DNA double helix is intriguing in one aspect as a pH dependent opening may affect the accessibility. Based on the suggestion of reviewer #1, point 9, we included an explanatory figure (Supplementary Fig. 11A) in to make this point clearer. We do not believe that actin filaments or microtubules are necessarily a good comparison as they do not share the basic antiparallel architecture and they tend to be more stable polymers easily discernible in electron microscopy images as opposed to the more flexible nature of the p62 filaments.

REVIEWERS' COMMENTS

Reviewer #1 (Remarks to the Author):

The authors performed extensive revisions which significantly improved the manuscript. The new proteomics data adds to the proposed working model of the p62/Ca²⁺ coat formation. Other concerns from this reviewers were also constructively addressed. As such, I would recommend the manuscript for publication in the current form.

I would still recommend spell check of the text and figures, e.g. Holliday junctions in one of the figures is spelled with one L.

Corrected as requested.

"In cellulo" should be spelled "in cellula" according to Latin grammar.

The term "in cellulo" is not genuine Latin, but has been coined in recent biology to mean experiments conducted within live cells (often in tissue culture or isolated single cells but not a whole organism) (<https://doi.org/10.1016/j.sbi.2024.102843>). To avoid any confusion, we revert to the more commonly used term "in situ" instead.

I've read the Reviewer 5 comments and responses by the authors and I think they are satisfactory. It is impossible for me to know if the Reviewer would accept them, but from the extensive experimental and textual revisions I would say most or all of the criticisms have been addressed. The authors did a very nice job in providing clarifications requested by the reviewer and formalising their model further. Undoubtedly, there are many uncertainties about the role of p62 and inorganic molecules at LDs but the data is robust, technology is cutting edge, and scientific advance and significance for the autophagy field and cell biology in general is obvious to me.

Reviewer #2 (Remarks to the Author):

The revised manuscript is fundamentally o.k.. It is in my opinion too long, however. But this is an editorial decision.

No changes required.

Reviewer #3 (Remarks to the Author):

No changes required.

Reviewer #4 (Remarks to the Author):

The revision substantially strengthens the manuscript and addresses most of my previous comments. I recommend acceptance pending minor revisions and one concern.

Concern:

1) The authors report a total dose of 500 e⁻/Å² for the EDX measurements. I am concerned that the dose might be too high, which caused too much damage to the sample.

Thank you for raising the concern. The reviewer may have compared the typical doses applied in cryo-ET with the reported total dose of 500 e⁻/Å² used for the EDX analysis. As reported in the manuscript (page 52): "EDX spectra were recorded with a 50 μs dwell time and 50 frames resulting in a total dose of 500 e⁻/Å²". Therefore, we were able to examine intermediate stages of the recording. Between 300 and 400 e⁻

$/\text{\AA}^2$, we observed initial sample bubbling and the resulting damage most likely compromised highest spatial resolution while the integrity of the ultrastructure (lipid droplet and electron-dense ring) was not affected. When we analyzed EDX spectra of the relevant areas from a total of $300\text{ e}^-/\text{\AA}^2$, we qualitatively identified the same elements as presented in Figure 6 and Supplementary Figures 7-9. Given this observation, we have no evidence for any dose-induced excessive mass loss that may affect the element identification. As EDX requires significant doses to detect element-specific signal, and therefore, add the following sentence to the Methods:

We found $500\text{ e}^-/\text{\AA}^2$, to be a good trade-off between high signal-to-noise ratio in the EDX spectra while still maintaining the ultrastructural integrity of the sample.

Minor revisions

2) Total tomography dose: Please recalculate and confirm the cumulative dose; the per-tilt values reported in Methods (lines 1030–1031 and 1233–1235) imply a total dose of $\sim 80\text{ e}^-/\text{\AA}^2$.

This was not an error. We added clarifying statements to both parts: the basic dose per tilt estimate is only valid for 0° tilt while it is increased for higher tilts as part of a SerialEM setting:

Each tilt was recorded as movie ... corresponding to a dose of approximately $2.0\text{ e}^-/\text{\AA}^2$ at 0° tilt...

All tilt series were recorded from -54° to 54° with 3° increments, using a dose-symmetric bidirectional acquisition scheme including 'the varied intensity' option (i.e. higher tilt angles received higher doses, as $1/\cosine$ of the tilt angle) in SerialEM.

3) Typo: Methods, "Cryo-electron tomography of reconstituted p62 filaments," line 5 (around line 1030) should read "63,000 \times magnification" (not "63.000 \times ").

Changed as requested.

4) Supplementary figure quality: Supplementary Figures 7–9 are more blurred in the current merged file than in the previous version. Please check the file format/resolution and restore the prior quality.

Thank you for pointing out this quality deterioration. We made sure to import high-resolution/quality images in the provided Supplementary information file.